

**The GEOVIDE cruise in May-June 2014 reveals an intense Meridional**
**Overturning Circulation over a cold and fresh subpolar North Atlantic**
Patricia Zunino[1], Pascale Lherminier[2], Herlé Mercier[1], Nathalie Daniault[3], Maria Isabel
García-Ibáñez[4] and Fiz F. Pérez[4]
[1] CNRS, Laboratoire d'Océanographie Physique et Spatiale (LOPS), IUEM, Plouzané, France.
[2] Ifremer, Laboratoire d'Océanographie Physique et Spatiale (LOPS), IUEM, Plouzané, France.
[3] Université de Bretagne Occidentale, Laboratoire d'Océanographie Physique et Spatiale (LOPS),
IUEM, Plouzané, France.
[4] Instituto de Investigaciones Marinas, IIM-CSIC, 36208 Vigo, Spain
Corresponding author: pzuninor@ifremer.fr
**Abstract**
The GEOVIDE cruise was carried out in the subpolar North Atlantic (SPNA), along the
OVIDE section and across the Labrador Sea, in May-June 2014. It was planned to clarify the
distribution of the trace elements and their isotopes in the SPNA as part of the GEOTRACES
international program. This paper focuses on the state of the circulation and distribution of
thermohaline properties during the cruise. In terms of circulation, the comparison with the
2002−2012 mean state shows a more intense Irminger current and also a weaker North
Atlantic Current, with a transfer of volume transport from its northern to its central branch.
However, those anomalies are compatible with the variability already observed along the
OVIDE section in the 2000s. In terms of properties, the surface waters of the eastern SPNA
were much colder and fresher than the averages over 2002−2012. Remarkably, in spite of
negative temperature anomalies in the surface waters, the heat transport across the OVIDE
section, estimated at $0.56 \pm 0.06$ PW, was the largest measured since 2002. This relatively
large value is related to the relatively strong Meridional Overturning Circulation measured
across the OVIDE section during GEOVIDE ($18.7 \pm 3.0$ Sv). Analyzing the air-sea heat and
freshwater fluxes over the eastern SPNA in relation to the heat and freshwater content
changes observed during 2013 and 2014, we concluded that these changes were mainly driven
by air-sea heat and freshwater fluxes rather than by ocean circulation.





## 1. Introduction

The subpolar North Atlantic (SPNA) is a key area for studying the effect of climate change in the ocean. The deep convection processes there behave as a driving mechanism for the Meridional Overturning Circulation (Kuhlbrodt et al., 2007; Rhein et al., 2011; Sarafanov et al., 2012), which transports heat to high latitudes in the North Atlantic and is predicted to slow down at the end of the present century (IPCC, 2007). Additionally, the SPNA presents the highest anthropogenic $CO_2$ storage rate of all oceans (Khatiwala et al., 2013), due to both the advection of surface waters enriched with anthropogenic $CO_2$ in the subtropical North Atlantic (Pérez et al., 2013; Zunino et al., 2015) and their deep injection in the subpolar gyre (Pérez et al., 2010). In addition, the SPNA is one of the few oceanic regions where significant cooling was detected over 1955–2010 while the rest of the world oceans was warming (Levitus et al., 2012). For all these reasons, the SPNA has been the target of several projects and broadly sampled by oceanographic cruises. As part of the OVIDE project (http://www.umr-lops.fr/Projets/Projets-actifs/OVIDE), the OVIDE section has been studied biennially in summer since 2002 to collect data related to the circulation and the carbon cycle. Its path between Greenland and Portugal is shown in Fig. 1 along with a schematic view of the upper, intermediate and deep circulations in the SPNA adapted from Daniault et al. (2016), which will be referred to as D2016 hereafter.

The international GEOTRACES program (http://www.geotraces.org/) aims to characterize the trace elements and their isotopes (TEIs) in the world ocean. These TEIs are Fe, Al, Zn, Mn, Cd, Cu, $\delta^{15}N$, $\delta^{13}C$, $^{231}Pa/^{230}Th$, Pb and Nd in the dissolved phase as well as in particles and aerosols. TEIs provide constraints and flux estimates that can be used to reconstruct the past environmental conditions. The GEOVIDE project is a French contribution to the GEOTRACES program. It is dedicated to measure the large-scale distributions of TEIs in the SPNA for the first time. The GEOVIDE cruise was carried out in May–June 2014 and was composed of two sections: one along the OVIDE line (its 7[th] repetition) and another one crossing the Labrador Sea, from Cape Farewell (Greenland) to St John's (Canada). The expertise gained on water mass properties and circulation across the OVIDE section (García-Ibañez et al., 2015; D2016) first helped to determine the optimal geographic distribution of the TEI sampling. However, the ocean is not steady, and the present study shows how anomalous the eastern SPNA was in summer 2014 compared with the previous decade, and thus provides guidance for the interpretation of the measured distribution of TEIs.



The ocean has uptaken 90% of the heat energy accumulated in the climate system since 1971 (Riser et al., 2016). In this context, it is striking to note the absence of a significant warming trend in between 50° N and 60° N in the Atlantic Ocean (Levitus et al., 2012; Sgubin et al., 2017). However, a strong variability occurs at the decadal timescale, with, in particular, warming and salinification of the SPNA detected from the mid-1990s to the mid-2000s (Bersch et al., 2007; Sarafanov et al., 2008). Some studies identified the North Atlantic Oscillation (NAO, Hurell et al., 1995) as a key atmospheric forcing explaining this variability. The reduction in the buoyancy-forced deep convection in the Labrador Sea was associated with the decline in the NAO index after 1996 and was identified as the cause of the observed warming, salinification and concurrent contraction/weakening of the subpolar gyre (Bersch, 2002; Häkkinen and Rhines, 2004; and Bersch et al., 2007). Robson et al. (2012) found that the rapid warming of the SPNA was primarily caused by durable northward ocean heat transport associated with the strengthening of the Meridional Overturning Circulation (MOC) in response to the increased surface buoyancy loss in the Labrador Sea during the prolonged positive NAO period in the late 1980s to early 1990s (see also Deshayes and Frankignoul, 2008; Lohmann et al., 2009; and Barrier et al., 2015). Other studies identified anomalies in the wind forcing in the inter-gyre gyre region as the cause of the 1995-1996 warming and salinification (Herbaut and Houssais, 2009; and Häkkinen et al., 2011).

Recently, the SPNA cooled and freshened again: Johnson et al. (2016) documented a SPNA region cooler in 2014 than in 1993-2014 climatology, this cooling intensified in 2015 and 2016 (Yashayaev and Loder, 2016; 2017). So, the GEOVIDE cruise crossed the SPNA region in a context that contrasts with the previous decade and could be the beginning of a new state. Over the eastern SPNA, Grist et al. (2015) analyzed the winter 2014 anomalous air-sea fluxes and their imprint on the ocean. Based on EN4 ocean reanalysis, they detected negative temperature anomalies in the surface waters, which they related to anomalous air-sea heat fluxes. Conversely, Holliday et al. (2015), who found evidence of similar cooling and also of freshening in the Irminger and Iceland basins from 2010–2011 to 2014, privileged the hypothesis of a remote source of those anomalies, i.e. the advection from the western SPNA. We will discuss both hypotheses in this study.

In this manuscript, we contextualize the physical background of the GEOVIDE cruise to help for the interpretation of distribution of TEIs in the eastern SPNA. Subsequently, by the analysis of the GEOVIDE cruise data along with altimetry, oceanic database and air-sea flux data, we disentangle the causes of the anomalous thermohaline properties of the surface and



intermediate layers of the eastern SPNA in May–June 2014. The paper is organized as
follows. Data and methodology are described in section 2. Section 3 displays the main results
on the large and mesoscale patterns of the circulation and thermohaline anomalies in 2014,
settling the GEOVIDE TEIs stations in this context. These results are discussed in section 4.
Finally, section 5 presents the main conclusions.

**2. Data and Methods**
**2.1. GEOVIDE data**
The GEOVIDE cruise was the French contribution to the GEOTRACES program
(http://www.geotraces.org/) in the North Atlantic. It was carried out on board the French R/V
"*Pourquoi Pas?*" from 15 May 2014 to 30 June 2014. A total of 78 stations were measured
and sampled along two hydrological sections: i) the 7[th] repetition of the OVIDE section (from
Portugal to Greenland) and ii) a section across the southern Labrador Sea, between Cape
Farewell and Newfoundland. In this paper we only deal with data from the OVIDE section.
Because this cruise was inserted in the GEOTRACES project, a large number of parameters
were measured, some of them in very low concentration. Therefore, several rosette casts (up
to 9) had to be done at some stations; the first cast was always used as reference for physical
characterization of water masses and currents. Stations were named according to the
parameters to be measured and the different number of casts to be carried out: Short, Large,
XLarge and Super stations. Nearly all the TEIs required by the GEOTRACES program were
sampled at Xlarge and Super stations, which positions were selected to be representative of
the different hydrographic regions, as detailed in section 3.4.
Because the ship time was limited to 45 days, the number of stations along the OVIDE section
was reduced compared with previous cruises, with 60 stations within 6 weeks during
GEOVIDE compared with 95 stations usually sampled within about 3 weeks in previous
OVIDE cruises. A sensitivity analysis was performed with the data from the 2010 OVIDE
cruise in order to select the station positions and minimize the error associated with the under-
sampling: as discussed later, the main water masses and currents crossing the OVIDE section
were correctly sampled during the GEOVIDE cruise. Conductivity, temperature, pressure and
dissolved oxygen were measured using a CTD SBE911 equipped with an SBE-43. The rosette
was also equipped with 22 bottles for collecting seawater. For calibration purposes, salinity
and oxygen were determined on board from seawater samples, using a salinometer and





titration, respectively. The final accuracy was 0.001°C, 0.002, and 2 µmol kg$^{-1}$ for
temperature, salinity and oxygen, respectively. Figure 2 shows the calibrated temperature,
salinity and oxygen measured during CTD-O$_2$ down casts of the OVIDE section. For more
details about the water mass properties and their distributions along the OVIDE section
between 2002 and 2012, see García-Ibañez et al. (2015) and D2016. Finally, the velocities of
the upper waters were measured continuously with two ship-mounted ADCP (Ocean
Surveyors) at a frequency of 38 Hz and 150 Hz, measuring down to 1000 m and 300 m, with
vertical resolutions of 24 m and 8 m, respectively.
The winter mixed layer depth (WMLD) was estimated along the OVIDE section by visual
inspection of the individual potential density and Apparent Oxygen Utilization (AOU)
profiles measured during the GEOVIDE cruise. Because the cruise was conducted in summer,
the seasonal mixed layer was disregarded and the WMLD was defined as the depth where the
slope of the density profile accentuated and the AOU was larger than 0.6 µmol kg$^{-1}$. The latter
value was chosen because it was the best fit with the density criteria at most stations.
**2.2. Inverse model**
The absolute geostrophic field orthogonal to the section was estimated by a box inverse model
using the hydrological profiles measured at each station, current measured by the ship
mounted ADCP and a volume conservation constraint of 1 Sv northward (Lherminier et al.,
2007). The inverse model is based on the thermal wind equation and the least-squares
formalism following the method described in Mercier et al. (1986) and Lux et al. (2001).
Additionally, the Ekman velocities were added to the inverse model: the Ekman transport was
estimated from NCEP winds (Kalnay et al., 1996) and equally distributed over the first 30 m.
The velocity errors were given by the resulting covariance matrix from the box inverse model.
For more details about the inverse model configuration specific to OVIDE, see Lherminier et
al. (2007, 2010) and Gourcuff et al. (2011). The volume transports were computed by
multiplying velocities by the distances between two stations. Their errors were obtained from
the full covariance matrix of velocities, taking into account error correlations, as explained in
Mercier (1986).
For the computation of transport across the OVIDE section from GEOVIDE data, the first
challenge was the spatial sub-sampling. In order to evaluate its consequences, the velocities
measured by the S-ADCP and those resulting from the inverse model are compared in Fig. 3
(note that the vertical scale differs between the subplots). We see that the inverse model
results reproduce the main features of the large-scale circulation captured by the S-ADCP. As





expected, mesoscale and ageostrophic structures of horizontal sizes smaller than the distances
between stations are visible on the S-ADCP section but are not resolved in the inverse model
solution (e.g. between stations 45 and 38 or between stations 32 and 27). However, because
the geostrophic velocity is an average between stations, this does not imply any bias in the
transports. This outcome is also supported by Gourcuff et al. (2011) who, comparing altimetry
and S-ADCP data, showed that the contributions of ageostrophic motions tend to cancel out
when averaged over the distance between stations.
The inverse model estimates the absolute geostrophic transport and the transport of heat and
other tracers. The under-sampling of the GEOVIDE cruise notably increases the errors
associated with the transport of tracers, because the horizontal gradients of those tracers are
less well resolved. The tracer considered in this work is temperature. By applying the
GEOVIDE subsampling to the inversion of the OVIDE 2010 data, we estimated a
supplementary and independent sampling error of 0.04 PW for heat transport.
**2.3. Oceanic database**
We used the In Situ Analysis System (ISAS) analysis (Gaillard et al., 2016), which, based on
Argo profiles and other qualified *in situ* observations (cruises, fixed-point time series, ships of
opportunity, etc.), produced monthly gridded fields of temperature and salinity profiles by
optimal interpolation for the period since 2002. We also used EN4 reanalysis. Similar to
ISAS, EN4 reanalysis is an optimal interpolation that incorporates *in situ* data measured since
1900, filling gaps by extrapolation from the observational data using covariances from the
Hadley Centre model (Good et al., 2013). We also used the temperature and salinity analysis
developed by JAMSTEC (Hosoda et al., 2008), which is also an optimal interpolation based
on Argo profiles, Triangle Trans-Ocean Buoy Network (TRITON) and other *in situ*
observations.
First, we evaluated the temporal and horizontal extension of the potential temperature ($\theta$) and
salinity (S) anomalies detected in the surface layer from ISAS: both properties were averaged
between 20 and 500 m at each ISAS grid point in the North Atlantic, and monthly anomalies
were then estimated with respect to the 2002−2012 mean values. Second, ISAS, EN4 and
JAMSTEC databases were used to evaluate the heat and freshwater content changes in the
upper 1000 m in the region delimited by 40°−60° N and 45°−10° W: for each month the heat
content ($HC_{month}$) and the freshwater content ($FWC_{month}$) of the volume of water in the box
previously defined was estimated following eq. 1/eq. 2:



$HC_{month} = \sum_{z=1}^{z=n} \sum_{i=1}^{i=n} \theta_{z,i} * Cp_{z,i}, \ \rho_{z,i} * V_{z,i}$          eq. 1
$FWC_{month} = \sum_{z=1}^{z=n} \sum_{i=1}^{i=n} \frac{(35 - S_{z,i}) * \rho_{z,i} * V_{z,i}}{35}$          eq. 2
where z and i are the depth levels and grid points of the database, and $Cp_{z,i}$, $\rho_{z,i}$ and $V_{z,i}$ are the
heat content capacity, density and volume of each depth level and grid point of the database.
**2.4. Air-sea flux data**
In order to evaluate the role of atmospheric forcing on the θ and S anomalies observed during
the GEOVIDE cruise, re-analyzed ERA-Interim data (Berrisford et al., 2011) and NCEP data
(Kanamitsu et al., 2002, http://www.esrl.noaa.gov/psd/) were processed. In particular, we
estimated seasonal anomalies of net air-sea heat flux (and its components: sensible heat, latent
heat, net longwave radiation and net shortwave radiation) and freshwater flux (and its
components: precipitation and evaporation) as follows. Firstly, seasonal means were
computed defining winter as DJF, spring as MAM, summer as JJA and autumn as SON.
Secondly, seasonal anomalies were calculated relative to the mean seasonal cycle of 2002–
2012. Finally, the anomalies of winter–spring 2014 that preceded the GEOVIDE cruise were
estimated.
Furthermore, the monthly time series of net air-sea heat and freshwater fluxes were used to
evaluate the contribution of the atmospheric forcing to the observed heat and freshwater
content changes in the box defined in section 2.3. Specifically, we integrated net air-sea heat
and freshwater fluxes from February 1, 2013 to December 31, 2014.

**3. Results**
**3.1. Circulation across the OVIDE section in 2014**
The OVIDE section is intersected by permanent currents and gyres that are described by
D2016 using the average measurements from the first 6 OVIDE cruises (2002 – 2012). This
section presents the intensity, location and extension of these dynamical structures during the
GEOVIDE cruise. The results showed hereafter are based on the solution of the inverse model
(see Fig. 3, lower panel). Despite the mesoscale structures typical of a single occupation of
the section, we can identify and quantify all the main patterns described by D2016.



Near Greenland, the Western Boundary Current (WBC) flows southwestward, guided by the
continental slope. During the GEOVIDE cruise, its extension towards the central Irminger Sea
at depths > 2000 m (see Fig. 3, lower panel) is marked by a bottom mesoscale feature typical
of the plume structure of the overflow (Spall and Price, 1997). The total intensity of the WBC
was estimated at $30.3 \pm 2.1$ Sv southward.
The cyclonic gyre defined as the Irminger Gyre (IG) by Väge et al. (2011) can be seen in the
western part of the central Irminger Sea. Following their definition, we quantified the
intensity of the IG by integrating the northward transport above the isotach 0 m s$^{-1}$ (Fig. 3b),
which amounted to $6.8 \pm 3.0$ Sv.
The Irminger Current (IC) flows northeastwards along the western flank of the Reykjanes
Ridge. In 2014, its top to bottom integrated transport amounted to $17.5 \pm 7.3$ Sv, which
accounts for both, the northward and the southward currents east of the IG. Considering only
the northward velocities brings the IC intensity to a value of $22.7 \pm 6.5$ Sv.
The Eastern Reykjanes Ridge Current (ERRC) flows southwestward east of the Reykjanes
Ridge. In 2014, its top-to-bottom integrated transport, between the Reykjanes Ridge and
station 34 (Fig. 3), amounted to $13.6 \pm 6.0$ Sv southward.
The North Atlantic Current (NAC) at the OVIDE section consists of meandering branches
flowing northeastward between the center of the Iceland Basin and the Azores-Biscay Rise
(D2016). To determine its horizontal extension, we used the barotropic streamfunction (Fig.
4) and AVISO altimetry data (Fig. 5). The NAC intensity was quantified as the accumulated
transport from the relative minimum of the barotropic streamfunction in the central Iceland
Basin up to the maximum of the barotropic streamfunction in the Western European Basin
(D2016). In the Iceland Basin, we found two relative minima of the streamfunction (Fig. 4)
due to the presence of an anticyclonic eddy, which was considered as part of the NAC, as
justified in the next section. The limits of the NAC along the OVIDE section are indicated by
green points in Fig. 5, between which the different branches of the NAC appear as energetic
northeastward currents. The top to bottom intensity of the NAC in 2014 amounted to $32.2 \pm$
11.4 Sv. Following D2016, three different branches of the NAC can be differentiated: the
northern branch, the subarctic front (SAF) and the southern branch. The SAF is identified as
the concomitant intense northward transport and salinity increase around 22.5° W (Fig. 4).  In
2014, top-to-bottom transport of the different NAC branches was $0 \pm 6$ Sv, $25 \pm 3$ Sv and $7 \pm$
5 Sv, respectively. Note that the northern branch of the NAC transport is null with a large



associated error and, by contrast, the SAF is remarkably large. This point is discussed in
section 4.
The easternmost dynamical feature of the OVIDE section is the NAC recirculation. Its
intensity of 10.1 ± 6.4 Sv southwestward is determined as the top-to-bottom accumulated
transport between the southern limit of the NAC and the easternmost station of the OVIDE
section.
The intensity of the Meridional Overturning Circulation (MOC) across the OVIDE section
was defined from the velocities given by the inverse model as the maximum of the surface to
bottom integrated streamfunction computed in vertical coordinates of potential density
referenced to 1000 m ($\sigma_1$). During the GEOVIDE cruise, it amounted to 18.7 ± 2.7 Sv and
was found at $\sigma_1 = 32.15$ kg m$^{-3}$. Additionally, using the independent monthly MOC index
created by Mercier et al. (2015), which is based on altimetry and Argo data, the intensity of
the MOC across the OVIDE section amounted to the compatible value of 21.3 ± 1.5 Sv in
June 2014, while the 2014 annual mean value of the MOC index was 18.2 Sv.
Heat transport during the GEOVIDE cruise was estimated at 0.56 ± 0.06 PW. Following the
Bryden and Imawaki (2001) methodology adapted by Mercier et al. (2015) in isopycnal
coordinates, we found 0.50 PW transported by the overturning circulation, 0.04 PW by the
horizontal or gyre circulation and 0.02 PW by the net transport across the section.
**3.2. Fronts and eddies**
Together with the above-mentioned permanent circulation features, we observed some
remarkable eddies during the GEOVIDE cruise that could modify the "typical" patterns of
properties defined by D2016 or García-Ibañez et al. (2015), as well as it can affect the
distribution of tracers measured during the GEOVIDE cruise.
The identification of eddies and fronts was based on the analysis of surface velocities
provided by AVISO (see Fig. 5), the velocity profiles given by both the S-ADCP and the
inverse model (Fig. 3) and the vertical distribution of properties (Fig. 2). In Fig. 5, we identify
clearly that the most energetic currents crossing the OVIDE section are the WBC, close to
Greenland, and the NAC with its different branches. Moreover, all the energetic eddies
intersecting the OVIDE section were observed in the NAC (Fig. 6) and identified on Fig. 3.
From north to south, the first eddy intersecting the section, referred to as the northern eddy, is
detected at 56.5° N, 27° W (Fig. 5). This eddy lies between stations 34 and 32 (Fig. 3; Fig. 6),



extending from the surface to the bottom but intensified in the upper 600 m. From Fig. 6, we inferred that this eddy was generated in April at approximately 56.5° N, 26° W from the meandering of the NAC north of the OVIDE section. In May 2014, the eddy was totally formed and intersected the section between 55.5° N and 57° N. In June 2014, the eddy moved southwestward, in agreement with the general displacement of anticyclonic eddies in the SPNA. The core of the northern eddy, between stations 34 and 32 in Figs. 2a and 2b, shows properties warmer and saltier than the surrounding water, confirming the NAC origin of this eddy; this is why this anticyclonic eddy has been considered as part of the northern branch of the NAC. Note that in May-June, the net transport of this eddy is almost 0 Sv (see Fig. 4 between stations 34 and 32).

A large anticyclonic eddy, the central eddy, is observed at 53° N, 26° W, at a tangent to the OVIDE section between stations 30 and 29 (red squares in Fig. 6). However, no signal was detected in the barotropic streamfunction (Fig. 4) since the northward and southward velocities (Fig. 3a) compensated once integrated between the two stations (Fig. 3b). It is noteworthy that, contrary to the previous anticyclonic eddy, this one is stationary south of the OVIDE section (see the monthly evolution in Fig. 6). Hydrographic properties measured at stations 29 and 30 showed cold and fresh water between 350 m and 500 m depth, typical of the Subarctic Intermediate Water (SAIW), which is most likely advected by this anticyclonic eddy.

The most remarkable front present on the OVIDE section is the SAF, associated with the central branch of the NAC. Along the OVIDE section, it is situated between 49.5° N and 51° N in latitude and 23.5° W and 22° W in longitude (Fig. 5, red points). This front separates cold and fresh water of subpolar origin from warm and salty water of subtropical origin; it is identifiable in Fig. 2 at station 26 by the steep slope of the isotherms and isohalines. The position of this front is known to vary spatially (Bersch 2002; Bower and Von Appen, 2008; Lherminier et al., 2010), creating anomalies of salinity and temperature that will be discussed later.

Finally, also in Fig. 5, we identified the southern branch of the NAC with a maximum in the eastward velocities found at 46.5° N, 22° W, just southwest of the OVIDE section. Despite a very rich mesoscale activity we can distinguish in Fig. 5 that the southern NAC splits into two sub-branches before crossing the OVIDE section, in agreement with D2016. The northernmost sub-branch cuts the section between stations 23 and 24 at 48.5° N, 21° W. The



southernmost sub-branch evolves into a cyclonic eddy (the southern cyclonic eddy) that
intersects the OVIDE section south of station 21. This eddy is also observed in the velocity
profiles (Fig. 3) between stations 21 and 19, as well as by the uplifting of isotherms and
isohalines in Fig. 2. To its southeast, an anticyclonic eddy, centered on station 18, marks the
southern limit of the NAC and the beginning of the southwestward recirculation. On the
OVIDE section, the southern anticyclonic eddy also marks the northwest limit of the presence
of Mediterranean Water at about 1000 m depth (Fig. 2b), consistently with its slow westward
advection since March (Fig. 6). Note that while the southern anticyclonic eddy looks stable
over time, the southern cyclonic eddy seems more transitory since it is not clearly visible in
April.
**3.3. Thermohaline anomalies in 2014**
The anomalies of potential temperature (θ), salinity (S) and dissolved oxygen along the
OVIDE section in 2014 were calculated relative to the 2002–2012 period (Fig. 7). Note that
the reference values were computed from six repetitions of the OVIDE section (summers
2002, 2004, 2006, 2008, 2010 and 2012) and only anomalies larger than one standard
deviation from the mean are represented in Fig. 7. In the following, S and θ anomalies were
quantified as the mean values of the anomaly patches represented in Fig. 7. We identified 3
different types of anomalies along the OVIDE section. First, negative anomalies in surface
waters were observed over the Reykjanes Ridge and east of 20° W. In the former, the S and θ
anomalies were quantified at -0.07 and -0.95°C, respectively. In the latter, the negative
anomalies of S and θ amounted to -0.11 and -0.70°C, respectively. The cooling and
freshening of the surface-intermediate waters were not compensated in density: the cooling
dominated and the water was significantly denser (Fig. not shown). Concurrently, a positive
oxygen anomaly was observed. All these anomalies are delimited at the bottom by the winter
mixed-layer depth (WMLD, orange line in Fig. 7).
In both the Irminger Sea and the Iceland Basin, positive anomalies of S and θ were observed
in waters deeper than 1000 m. In the Irminger Sea, the S and θ anomalies amounted to 0.017
and 0.122°C, respectively. In the Iceland Basin, they reached similar values, i.e. 0.014 and
0.125°C. In both basins, these anomalies coincided with significant negative oxygen
anomalies up to -20 μmol kg$^{-1}$, suggesting that this water mass was not recently ventilated.



In the Iberian Abyssal Plain (IAP), negative anomalies of S (-0.12) and θ (-0.67°C) were observed at the level of the Mediterranean Water (MW), above and below the isopycnal 32.15 kg m$^{-3}$. Although remarkable, those anomalies are difficult to interpret because of the high variability of the Meddy distribution in this area.

The displacement of fronts or eddies already identified in the previous section generated other occasional anomalies. The salty and warm anomaly found at 27.4° W, above isopycnal 32.15 kg m$^{-3}$, is explained by the anticyclonic eddy (the northern eddy), which advected water from the NAC. The fresh and cold anomaly localized at 25° W is a consequence of the SAIW brought by the anticyclonic eddy (the central eddy) located at 53° N, 26° W and touching the OVIDE section between stations 30 and 29. Finally, the southeastward displacement of the SAF created a fresh and cold anomaly between 23° W and 22° W because warm and salty North Atlantic Central Water (NACW) usually found in this area was replaced by subpolar water.

Zooming out (Fig. 7), we found an increase in the ventilation in the first 1000 m, while the deeper waters are less oxygenated when compared to the 2002 – 2012 period. Remarkably, the oxygen anomalies are anti-correlated with the θ-S anomalies.

### 3.4. Settling the special GEOVIDE stations in the framework of the large-scale and mesoscale circulation

As part of the GEOTRACE program, seven superstations and three Xlarge stations were carried out along the OVIDE section in 2014. Here, we contextualize the superstations and Xlarge stations (red and green numbers, respectively, in Figs. 2, 3 and 4, and pink stars in Fig. 5) in the physical framework described above. Apart from station 26, which was specifically selected in real-time in the middle of the SAF, and station 38 over the Reykjanes Ridge, all the other special stations are representative of relatively large hydrographic domains since they are not strongly affected by the peculiar mesoscale features described in section 3.2.

Specifically, from Greenland to Portugal, these stations were located in: the East Greenland Coastal Current (EGCC, station 53), the East Greenland-Irminger Current (EGIC, station 60, same position than 51), the Irminger Gyre (station 44, same position than station 46), in the middle of the Iceland Basin (being part of the NAC northern branch, station 32), in the NAC southern branch (station 21), in the center of the southward recirculation in the IAP (station 13), on the Iberian Peninsula slope (station 8) and, finally, on the Portuguese continental shelf





(station 2). Importantly for the GEOTRACES community, although the superstations and
XLarge stations are representative in terms of circulation, the large-scale S − θ anomalies
detailed in section 3.4 need to be taken into account when comparing GEOVIDE data with
data from the previous decade.

**4. Discussion**
**4.1. State of the circulation during the GEOVIDE cruise in relation to the mean state**
We will first discuss the circulation patterns seen during the GEOVIDE cruise in comparison
with the mean position, extension and intensity of the main currents intersecting the OVIDE
section defined by D2016. Despite the coarse resolution of the GEOVIDE stations, all the
circulation structures are identified in the inverse model solution (Table 1). The intensity of
the WBC and the IG are similar to the mean state with a quite high reliability (low relative
error). The transports of the ERRC and the southwestward recirculation in the IAP are also
very similar to the mean state, but remained to a large degree uncertain. Conversely, the IC
and NAC are different from the mean state, but not significantly.
To go further in the analysis of IC, we compared its northward component near Reykjanes
Ridge with its equivalent from the 2002–2012 mean data (not shown in D2016). In this case,
the IC amounted to 22.7 ± 6.5 Sv, which is significantly larger than the northward IC
computed from D2016 data: 11 ± 3.4 Sv. Our result is similar to the estimate by Väge et al.
(2011) who quantified the IC at 19 ± 3 Sv (1991–2008). Therefore, we conclude that the thus-
defined IC was strengthened in 2014 in relation to the 2002–2012 mean value. Note that the
northward component of the IC, between stations 38 and 41, transports water masses that are
warmer and saltier than those advected southward, between stations 41 and 45, (Fig. 2); so the
intensification of the Irminger Current is meaningful in terms of transport of warm and salty
water to the north, and actually contributes to the upper limb of the MOC (Fig. 4, dotted line).
Concerning the weaker NAC intensity in 2014, it is very likely that the difference comes from
the change in the intensity of the northern branch of the NAC: 0 ± 6 Sv was computed in
GEOVIDE, while 11 ± 3 Sv was estimated by D2016. We believe that the weakening of the
northern branch of the NAC in 2014 was due the high mesoscale activity along the Maury
Channel in the Iceland Basin (Fig. 5), with anticyclonic eddies flowing southwestward that
temporarily blocked the northeastward propagation of the northern branch of the NAC. It is





possible that part of the current was deflected westward into the intensified Irminger Current.
However, we noticed that the intensity of the central branch of the NAC simultaneously
nearly doubled in 2014 compared with the 2002–2012 mean (25 ± 3 Sv vs. 14 ± 6 Sv),
suggesting there was also a partial transfer of transport from the northern to the central branch
of the NAC.
The SAF, that bears the central branch of the NAC, shows also a remarkable southeastward
displacement in 2014 in relation to the mean circulation pattern (Fig. 1), of about 100 km. In
March 2014, Grist et al. (2015) also detected a southward displacement of the NAC along the
30° W meridian by the analysis of EN4 data. However, it should be noted that their result
concerns a more southern branch of the NAC (41° N) that does not cross the OVIDE section
and recirculates southward in the Azores Current (Fig. 1).
Moreover, D2016 also defined some permanent circulation features where the velocity was
found to be in the same direction for all repeated measures on the OVIDE section 2002–2012
(see their Fig. 4). In our Fig. 3, we found most of these permanent circulation features: the
WBC, IC, ERRC, two deep southward veins transporting the ISOW in the Iceland Basin, and
the northward transport over Eriador Seamount in the intermediate layer. Only the
"permanent" anticyclonic eddy marking the southern limit of the NAC moved: it was
expected between station 20 and 21 according to the mean circulation (Fig. 1), but was instead
found at station 18, i.e. more to the southeast, during the GEOVIDE cruise (and called the
southern anticyclonic eddy previously).
The inverse model solution also provides a robust estimate of both the intensity of the MOC
and the heat transport. We observed a heat transport of 0.56 ± 0.06 PW. To compare it with
the 2002–2010 average, we used the data of Mercier et al. (2015), without data from 1997,
and obtained 0.47 ± 0.05 PW. Even if the 2014 value is not statistically different from the
mean, it is surprising to find such a high heat transport considering the cold anomaly observed
in the NAC surface waters (Fig. 7). To determine the role of the MOC in this result, we first
looked at the 2014 MOC (18.7 ± 2.7 Sv), which is 2.5 Sv higher than the 2002–2010 average
(16.2 ± 2.4 Sv). Note that including 2012 data (15 Sv and 0.39 PW, not published) in the
mean increases the difference with 2014. To improve our quantification of the influence of the
MOC on heat transport, we used the heat transport proxy HF* built by Mercier et al. (2015),
which evaluates the heat transport only driven by the diapycnal circulation, known to be the
dominant term of heat transport for all the OVIDE cruises. The proxy (eq. 3) is based on the



MOC intensity ($MOC_\sigma$) and the temperature difference between the upper and lower limbs of
the MOC ($\Delta T$):
$$HT^* = \rho.c_p.\Delta T.MOC_\sigma \qquad \text{(eq. 3)}$$
where $HT^*$, $\rho$ and $c_p$ are the heat transport proxy, the *in situ* density and the specific heat
capacity, respectively. During GEOVIDE, $HT^*$ amounted to 0.49 PW, with $MOC_\sigma = 18.7$ Sv
and $\Delta T = 6.40°C$. The 2002–2010 mean values of $HT^*$, $MOC_\sigma$ and $\Delta T$ were 0.43 PW, 16.2 Sv
and 6.79°C, respectively. So, the heat transport index and $MOC_\sigma$ were larger in 2014 than the
mean values, while the $\Delta T$ was smaller, which is consistent with the cold anomaly. These
results show that the larger $MOC_\sigma$ measured during GEOVIDE was enough to compensate for
the heat transport decrease due to the cooling of the surface waters. This result contrasts with
the study of Desbruyères et al. (2015), who argued that the long-term variability of the ocean
heat transport at the OVIDE section is dominated by the advection by the mean velocity field
of temperature anomalies formed upstream rather than the velocity anomalies acting on
temperature.

**4.2. Negative anomalies of θ and S in surface-intermediate layers explained by the local atmospheric forcing.**

The negative anomalies of θ and S in the surface-intermediate layers along the OVIDE
section in May–June 2014 were actually present over the whole of the year 2014 and the
whole SPNA (Fig. 8). θ and S anomalies in the ocean can be caused by changes in the lateral
advection of water masses with different properties, and/or by anomalous net air-sea fluxes.
The mean winter–spring (W-S 2014) anomalies of air-sea heat flux presented strong negative
anomalies over the whole SPNA (Fig. 9a), i.e. the ocean lost more heat than usual with
contribution of sensible and latent air-sea heat fluxes (Fig. 9b and 9c). The high latent heat
loss is associated with high evaporation, which can be seen in Fig. 9e. The net freshwater gain
(Fig. 9d) shows that high precipitation rates (Fig. 9f) overcame the freshwater loss by
evaporation. These anomalous air-sea heat and freshwater fluxes in the eastern SPNA suggest
that the negative θ and S anomalies observed in the surface-intermediate waters during
GEOVIDE were formed locally by atmospheric forcing.
The heat/freshwater content changes in the upper 1000 m of the ocean during the 2013–2014
period were evaluated together with the air-sea heat/freshwater fluxes in a region in the
eastern-SPNA delimited by 40–60° N latitude and 45–10° W longitude. In agreement with



Grist et al. (2015), we found that the air-sea heat flux is the main responsible for the cooling observed in the surface-intermediate layers. Exactly, we estimated the accumulated air-sea heat loss from summer 2013 to summer 2014 at $6.8 \ 10^{21}$ J, while the accumulated ocean heat loss for the same period amounted to $4.8 \ 10^{21}$ J (averaged of ISAS, EN4 and JAMSTEC estimates). Moreover, we detected that, despite the variability in freshwater content change at intra-seasonal and seasonal timescales (Fig. 10), there is a good agreement between the trends shown by the ocean freshwater content and the air-sea freshwater flux over the 2013–2014 period. These results support our conclusion that the negative θ and S anomalies observed in the surface-intermediate waters during the GEOVIDE cruise were locally formed by atmospheric forcing. The dominant role of the air-sea heat flux over the changes of ocean heat content contrasts with the results of several studies that showed that the heat content variability in the SPNA is mainly controlled by oceanic heat transport variability (e.g. Hátún et al., 2005; Marsh et al., 2008; Desbruyères et al., 2015).

More evidence for the important role of air-sea fluxes is provided by the distribution of θ, S and oxygen anomalies in the water column. Indeed, the WMLD along the OVIDE section east of the Reykjanes Ridge coincided with the deep limit of the anomalies most of the time (Fig. 7). The sign of the anomalies is consistent with vertical mixing in the winter before the GEOVIDE cruise, transferring the cold, fresh and oxygenated anomalies imprinted locally by the atmosphere into the whole mixed layer. Remarkably, the orange line in Fig. 7 reaches 1200 m in the Irminger Sea while deep convection did not exceed 700 m in winter 2014 in the central Irminger Sea (Duchez et al., 2016; Piron, 2015). It most likely results from the advection in the depth range 700–1200 m of high-oxygen intermediate water with densities slightly denser than the water above and possibly formed south of Greenland as suggested by Fig. 5.3 of Piron (2015).

Below the orange line in Fig. 7, we observed mainly warming, salinification and deoxygenation. This is in agreement with the tendencies observed since 2002 along the OVIDE section. Deep waters below 1300 m depth in the Irminger Sea were obviously not recently renewed, apart from the plume of DSOW. Kieke and Yashayaev (2015) showed the evolution of S and θ in the LSW measured in the Labrador Sea: below 1300 m, the positive tendencies of S and θ were similar to those observed in the Irminger Sea, and concerned the dense LSW formed in the 1990s.



Negative S anomalies of the surface waters of the SPNA were observed in the 1970s, during
the Great Salinity anomaly event, and were explained by a larger pulse of freshwater getting
into the SPNA through the Denmark Strait (Dickson et al. 1988; Robson et al., 2014).
Concurrently, the SPG started a cool phase that persisted up to the beginning of the 1990s and
was explained by the decrease in the ocean heat transport convergence with a minor
contribution of atmospheric forcing (Williams et al., 2014; Robson et al., 2014). Later, from
mid-1990s to mid-2000s, positive anomalies of θ and S in the surface waters of the SPNA
were observed, coinciding with the contraction and weakening of the SPG (e.g. Bersch, 2002;
2007; Sarafanov et al., 2008; Häkkinen et al., 2011). In the introduction, we detailed the
different hypotheses postulated by different authors to explain these anomalies, all of whom
interpreted the anomalies as originating in the Labrador Sea. Similarly, Hermanson et al.
(2014), by analyzing three versions of the Met Office Decadal Prediction System, identified
the three periods of cooling-warming of the SPNA indicated above: cooling from the
beginning of the 1970s, warming from mid-1990s to mid-2000s, and cooling from 2014, with
the latter predicted to continue at least up to 2017 and recently confirmed by data (Piron et al.,
2017; Yashayaev and Loder, 2017). For these three events, the authors found that the
mechanism controlling the anomalies was the heat convergence related to changes in MOC
intensity.
The 2014 anomaly was the first detected, after approximately 18 years of warming and
salinification. The winter NAO index for winter 2014 was positive and high (0.92), so,
following Bersch et al. (2007), an expansion of the subpolar gyre (SPG) would be expected.
Although we observed a southward displacement of the SAF in 2014, we could not prove the
link between the probable expansion of the SPG and the advection of additional subpolar
water northeastward. By contrast, we showed that the cooling and freshening of the surface-
intermediate waters observed in summer 2014 were locally formed in the eastern SPNA by
the atmospheric forcing.

**5. Summary and conclusions**
This paper addresses two main issues: first, under the umbrella of the GEOTRACES program,
it contextualizes the physical background of the GEOVIDE cruise carried out in May–June
2014, which is essential for the interpretation of distribution of TEIs in the eastern SPNA.



Second, it elucidates the cause of the cold and fresh anomaly detected in the surface waters of
the eastern SPNA in May–June 2014.
Concerning the circulation across the OVIDE sections, the most important difference between
the GEOVIDE state and the 2002–2012 mean state defined by D2016 is a strengthened
Irminger Current and a weaker North Atlantic Current, with a possible transfer of volume
transport from its northern branch to both its central branch and the Irminger Current. The
intensity of the MOC was the highest measured at the OVIDE section since 2002, $18.7 \pm 3.0$
Sv, and was high enough to compensate the negative temperature anomaly detected in the
surface waters, resulting in a high heat transport across the OVIDE section, $0.56 \pm 0.06$ PW.
The special GEOVIDE stations where the trace elements were measured were indeed
representative of the targeted hydrological regions, away from the core of the main advected
eddies identified along the sections. Nevertheless some precautions should be taken when
comparing with previous years since temperature, salinity and oxygen of the SPNA winter
mixed layer were significantly different from the 2002–2012 mean.
Finally, we demonstrated that the cold and fresh anomalies in the 2014 mixed layer induced
consistent changes in heat and freshwater content of the SPNA, and that they were driven by
atmospheric forcing. Our results elucidate the important role of air-sea flux in the θ-S changes
in this region, overcoming the warming and salinification induced by the increase in the MOC
amplitude and associated heat transport.

**Acknowledgements**
We gratefully acknowledge the crew of the *Pourquoi Pas*? vessel for their help and assistance
during the cruise and for winch repairs. We also acknowledge the work of the UTM-CSIC
(Spain) technical staff for the CTD manipulation. The GEOVIDE cruise would not have been
achieved without the technical skills and the commitment of Catherine Kermabon, Pierre
Branellec, Philippe Le Bot, Olivier Ménage, Stéphane Leizour, Michel Hamon and Floriane
Desprez de Gésincourt (LOPS) and also Fabien Pérault and Emmanuel de Saint Léger
(CNRS). The NCEP Reanalysis 2 data were provided by the NOAA/OAR/ESRL PSD,
Boulder, Colorado, USA, from their web site at http://www.esrl.noaa.gov/psd/. The altimeter
products were produced by Ssalto/Duacs and distributed by Aviso with support from CNES.
For this work, P. Zunino was supported by CNRS and IFREMER, within the framework of
the projects AtlantOS (H2020-633211) and GEOVIDE (ANR-13-BS06-0014-02),
respectively. H. Mercier was financed by CNRS, P. Lherminier by Ifremer and N. Daniault by
the University of Western Brittany, Brest. M.I. García-Ibáñez and F.F. Pérez were supported
by the Spanish Ministry of Economy and Competitiveness through the BOCATS (CTM2013-




41048-P) project co-funded by the Fondo Europeo de Desarrollo Regional 2007−2012
(FEDER).

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

**TABLES**
Table 1. Intensity (top-to-bottom integrated) of the different dynamical structures defined in
section 3.1 for 2014 and the mean values (2002-2012) estimated by Daniault et al. (2016).

| Units: Sv | WBC | IG | IC | ERRC | NAC | Recirculation |
|---|---|---|---|---|---|---|
| GEOVIDE | -30.3 ± 2.1 | 6.8 ± 3.0 | 17.5 ± 7.3 | -13.6 ± 6.0 | 32.2 ± 11.4 | -10.2 ± 6.4 |
| MEAN (2002-2012) | -33.1 ± 2.6 | 7.7 ± 2.1 | 9.5 ± 3.4 | -12.1 ± 1.1 | 41.8 ± 3.7 | -13.0 ± 2.0 |







**FIGURE CAPTIONS**
**Fig. 1.** Schematic diagram of the large-scale circulation adapted from Daniault et al. (2016).
Bathymetry is plotted in color with color changes at 100 m, 1000 m and every 1000 m below
1000 m. The locations of the GEOVIDE hydrographic stations are indicated by black dots
along the OVIDE section and across the Labrador Sea. Red dots, and associated numbers,
along the OVIDE section show the stations delimiting the regions used in this paper for the
transport computations. Superstations and XL stations carried out during GEOVIDE are
represented by pink stars. The main topographical features of the Subpolar North Atlantic are
labeled: Azores-Biscay Rise (ABR), Bight Fracture Zone (BFZ), Charlie–Gibbs Fracture
Zone (CGFZ), Faraday Fracture Zone (FFZ), Maxwell Fracture Zone (MFZ), Mid-Atlantic
Ridge (MAR), Iberian Abyssal Plain (IAP), Northwest Corner (NWC), Rockall Trough (RT),
Rockall Plateau (Rockall P.) and Maury Channel (MC). The main water masses are indicated:
Denmark Strait Overflow Water (DSOW), Iceland–Scotland Overflow Water (ISOW),
Labrador Sea Water (LSW), Mediterranean Water (MW), and Lower North East Atlantic
Deep Water (LNEADW).

**Fig. 2**. Vertical section of potential temperature (°C), salinity and oxygen (µmol kg$^{-1}$) along
the OVIDE section measured during the GEOVIDE cruise. The horizontal grey lines in the
three plots represent the isopycnal layers ($\sigma_1$ = 32.15 kg m$^{-3}$, $\sigma_2$ = 36.94 kg m$^{-3}$, $\sigma_4$ = 45.85 kg
m$^{-3}$) indicated in the upper plot. The vertical grey lines in the three plots are the limits
between the different circulation components crossing the OVIDE section: Western Boundary
Current (WBC), Irminger Gyre (IG), Irminger Current (IC), Eastern Reykjanes Ridge Current
(ERRC), northern branch of the North Atlantic Current (NNAC), SubArtic Front (SAF),
southern branch of the North Atlantic Current (SNAC) and the recirculation in the Iberian
Abyssal Plain (RECIR.). The main water masses are indicated in the central plot: Denmark
Strait Overflow Water (DSOW), Iceland–Scotland Overflow Water (ISOW), Labrador Sea
Water (LSW), Sub-Polar Mode Water (SPMW), Sub-Arctic Intermediate Water (SAIW),
North Atlantic Central Water (NACW), Mediterranean Water (MW) and North East Atlantic
Deep Water (NEADW). The main topographic features are indicated in the bottom plot:
Reykjanes Ridge (RR), Eriador Seamount (ESM), Western European Basin (WEB), Azores-
Biscay Rise (ABR) and Iberian Abyssal Plain (ABP). Ticks at the top of the upper and central
plots indicate the positions of all the stations measured during GEOVIDE, along the OVIDE



section, with some station numbers given above. In the bottom plot, the red and green
numbers indicate the position of the superstations and XLarge stations, respectively.

**Fig. 3**. Velocities (m s$^{-1}$) orthogonal to the OVIDE section measured during the GEOVIDE
cruise. Positive/negative values indicate northeastward/southwestward velocities. a)
Velocities measured by the ship-ADCP. b) Geostrophic velocity obtained by the inversion
model plus Ekman velocities in the upper 30 m. The vertical black lines are the limits between
the different circulation components crossing the OVIDE section as defined in the main text
and at the bottom of Fig. 2a. The horizontal discontinuous black line delimits the 800 dbar for
comparison of Fig. 3a and 3b. The horizontal black continuous lines are the isopycnals $\sigma_1 =$
$32.15 \, \mathrm{kg \, m^{-3}}$, $\sigma_2 = 36.94 \, \mathrm{kg \, m^{-3}}$ and $\sigma_4 = 45.85 \, \mathrm{kg \, m^{-3}}$. Bold numbers inside the figure are the
volume transports (in Sv) estimated for each region and vertical layer, with errors in
parentheses. The only exception is the estimation of the IG transport, which, following Väge
et al. (2011) was computed as the northward transport (the 0 m s$^{-1}$ isotach is indicated as a
thin black line in Fig. 3b in the western Irminger Sea). Station numbers at the top of the figure
are color-coded: black for regular stations, blue for large stations, green for XLarge stations
and red for superstations. The eddies described in section 3.2 are indicated at the top of the
plots.

**Fig. 4**. Streamfunction or volume transport horizontally accumulated from Greenland to each
GEOVIDE station, down to Portugal, and vertically accumulated in the upper limb of the
MOC (red discontinuous line) and in the whole water column (red continuous line). The mean
salinity in the upper limb of the MOC is also shown by the blue line and labeled on the right-
hand axis. Acronyms in the top of the figure indicate the different components of the
circulation crossing the OVIDE section as defined in Fig. 2. See Fig. 3 for station numbers
and bathymetry legend.

**Fig. 5.** Surface velocities (m s$^{-1}$) derived from AVISO data: arrows indicate current direction
and colors indicate current intensity. The white line represents the OVIDE section. The red
and green points indicate the extension of the different dynamical structures crossing the
OVIDE section in 2014. The green points delimit the extension of the NAC. The pink stars





indicate the position of the GEOVIDE superstations and XLarge stations. The bathymetry
contours, every 1000 m, are indicated by light grey lines.

**Fig. 6**. Surface velocities derived from AVISO data, as in Fig. 5 but zooming in on the NAC
region in March 2014, April 2014, May 2014 and June 2014. The yellow, red, clear green and
orange squares indicate the position of the northern, central and southern eddies, respectively,
discussed in section 3.2. The numbers of the GEOVIDE stations are indicated in all the plots:
pink for the superstations and Xlarge stations, and yellow for regular stations. The red and
green points delimitate the position of the SAF and the NAC, respectively, at the period of the
GEOVIDE cruise.

**Fig. 7**. Anomalies of potential temperature (upper panel, in °C), salinity (middle panel) and
oxygen (bottom panel, in $\mu$mol kg$^{-1}$) in 2014 in relation to the OVIDE 2002–2012 mean.
Only anomalies larger than one standard deviation of the 2002–2012 values are colored in the
figure. Station numbers follow the color code of Fig. 2. The orange line indicates the winter
mixed-layer depth (WMLD); in the Irminger Sea, the dotted line indicates the WMLD that
was not formed locally (see 4.2). The acronyms in the bottom plot are as in Figs. 2 and 3.

**Fig. 8.** Annual mean anomalies of potential temperature (left panel) and salinity (right panel)
in the surface waters (20–500 m) in the North Atlantic, estimated from ISAS database. The
reference period for estimating the anomalies was 2002–2012. The OVIDE section is
represented by a black line. Only anomalies larger than one standard deviation are colored in
the figure.

**Fig. 9**. Winter–Spring (DJFMAM) mean anomalies. The anomalies were calculated in
relation to the period 2002–2012. A, B and C are the total heat, sensible heat and latent heat
air-sea flux, respectively, in W m$^{-2}$; positive/negative values indicate ocean heat gain/lost. D,
E and F are net gain of freshwater, evaporation and precipitation; the unit is 10$^{-4}$ m;
positive/negative values indicate ocean freshwater gain/loss. The contours of anomalies 0 W
m$^{-2}$ (in a, b and c) and of 0 m (in d, e and f) are represented by a white line. Data source:



ERA-Interim. The green square represents the area for which the changes of heat/freshwater
content, and the integrated air-sea heat/freshwater flux represented in Fig. 10 were evaluated.

**Fig. 10.** Time series of the accumulated freshwater content change (in $m^3$) since February
2013 in the upper 1000 m of the box delimited by $40° - 60°$ N and $45° - 10°$ W computed from
three datasets: EN4 (blue), ISAS (red) and JAMSTEC (green). Accumulated air-sea flux
anomalies over the same box are also plotted in black and were converted into the same unit
by repartition in the box volume; data source: ERA-Interim.

Figure 1



Fig. 2



Fig. 3

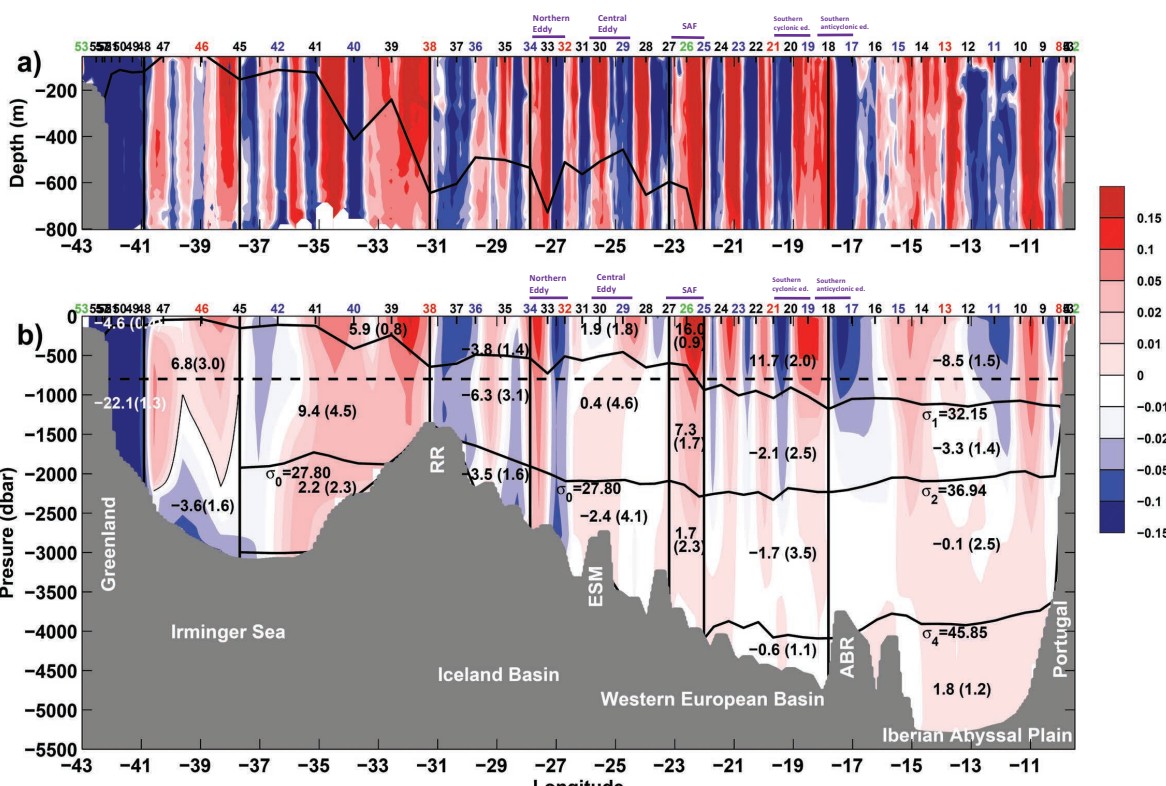



Fig. 4



Fig. 5

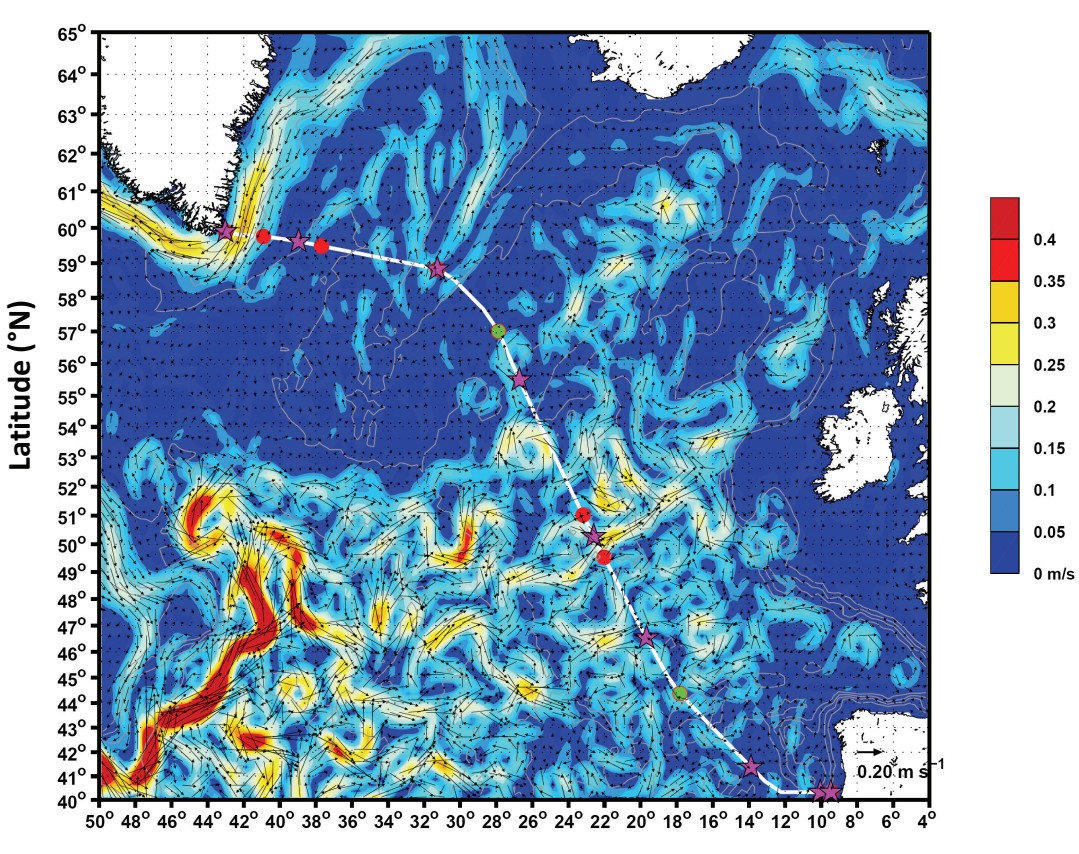





Fig. 6

Fig. 7






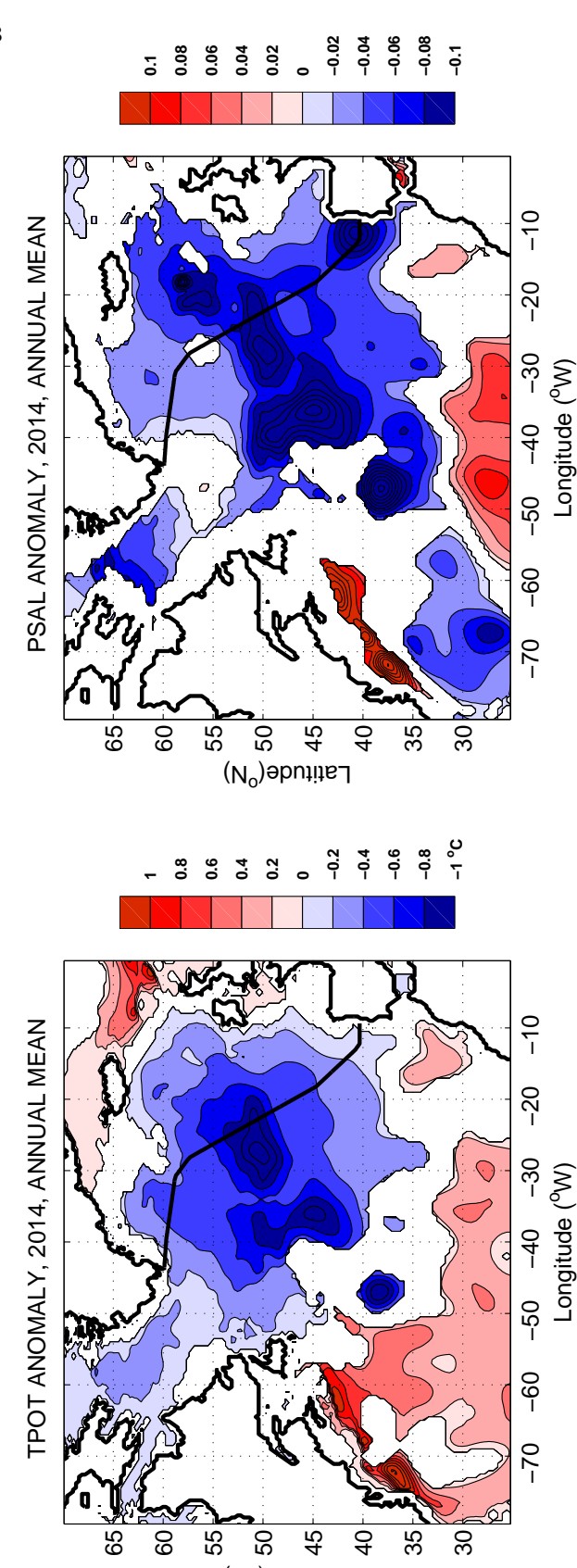

Fig. 8





Fig. 9





Fig. 10

