# Peer review of "The GEOVIDE cruise in May-June 2014 reveals an intense Meridional"

_Biogeosciences, 2017_

## Referee Comment (RC1) · Anonymous Referee #1 · 15 Aug 2017

One of the main findings in this paper is that despite the ongoing cooling/freshening anomaly in the SPNA, the authors measure stronger heat transport across the OVIDE section during the GEOVIDE cruise in May-June 2014. The authors further conclude by strongly stating that air-sea (heat and freshwater) fluxes were the dominant factor for the observed changes and that ocean circulation played a minor role.

The authors seem to have ignored the fact that the subpolar gyre (SPG) cooling/freshening is part of a decadal trend most likely initiated by ocean advection and circulation. There are evidence that the SPG temperature and salinity reversed already in 2005 (e.g. Robson et al. 2016, Nature Geoscience), and that the NAO shifted more

to the positive state later on around 2011. The 2014 cooling/freshening thus may have been enhanced by air-sea fluxes (not locally formed), but it is important to keep in mind that the cooling as well as the freshening did not start in 2014 as the authors mention. It was well underway!

Nevertheless, I think the paper is good and taking into account what I have mentioned above as well as my comments below will help improve the paper, which I recommend for publication after these issues have been resolved.

Major comments

I think the authors need to focus on the 2014 event as being part of the (multi-)decadal cooling/freshening rather than the instigator of it. A robust discussion along these lines is therefore strongly recommended.

Furthermore, I wonder also how the authors reconcile the fact that there are numerous studies (see e.g. Robson et al. 2017, Clim Dyn, and references therein) demonstrating that the western SPG is dominated by surface fluxes, while the eastern SPG is dominated by ocean advection.

How do you explain the large-scale salinity anomaly in Fig. 8 that spans both the the SPNA and the region of the Gulf Stream? there is very little discussed about this basin-scale feature.

Comments

I think the calculations and results are straightforward, but I have some comments that can help to further improve the paper:

Using the inverse model the authors and thereby identify all main flows, and show that the ship-ADCP velocities are largely similar, at least in structure, to those based on the inverse modelling. I suggest to compare your transport numbers, whenever possible, to other studies for a complete picture.

Weaker NAC during 2014 is an interesting finding. The top-to-bottom transport of 0±6Sv as compared to the 11±4 Sv estimated by D2016 is large. I believe it is possible to show this large-scale shift from altimetry along the eddy-blockage and the doubling of the intensity of the SAF, which the authors are briefly mentioning in lines 404-412. Please elaborate also on the transfer of transport from the northern to the central branch, it is not clear to me how this occur!

Furthermore, as the authors mention (lines 520-527), they expected an expansion of the SPG. Could you better discuss what the displacement of the SAF actually means?

The green box in Fig. 9 seems very large to me to be considered as the eastern North Atlantic. It is rather peculiar that in the net freshwater field you are averaging over an area that almost symmetrically includes positive and negative net FW.

The eddy part of the paper is clear, although full of details, it completes the picture well. It is however not easy from a visualization point view to see the eddies and the colors in the figure. Suggest to improve this and make it as clear as possible to the readers, perhaps similar to Fig. 6 of D2016.

Consider adding a reference to a recent paper by Rossby et al. (2017, JGR) on the fluxes across 59.5N. Their MOC transport estimate is in line with yours.

Minor comments

Please replace 'Hydrological' with 'Hydrographic' throughout.

Figure 1 is busy and therefore making an effort to explain all the signs is important. For example, you should indicate what the stars represent early on. You may also want to add the names of the different NAC branches here.

lines 251: Please keep it consistent with the decimal throughout the paper.

The bathymetry can hardly be seen in the AVISO figures.

lines 505: Define the SPG acronym. And no need for the SPG acronym in the last

paragraph of the discussion.

Caption Fig. 9; It is not clear in the text that the anomalies are for 2014.

---

## Referee Comment (RC2) · Anonymous Referee #2 · 16 Aug 2017

The GEOVIDE cruise in May–June 2014 reveals an intense Meridional Overturning Circulation over a cold and fresh subpolar North Atlantic, by Patricia Zunino et al.

This paper discusses the physical background of the GEOVIDE cruise in 2014. It highlight changes in transport as well as heat and freshwater content compared to the 2002-2012 mean state. The most interesting conclusion is that the large scale cooling seen in the SPG is (more then) compensated by a strengthened circulation in the net heat transport.

One comment on the structure of the paper. Although the TEI measurements do not feature in the abstract there are discussed several time in the introduction and elsewhere in the paper. It therefore reads like the discussion of the TEI measurements will be discussed later in the paper, but this never happens. It is also never mentioned where (in which other paper) these measurements will be shown. This does not improve the overall clarity of the paper. The authors should rephrase the text as to make it more clear what the focus of this paper is, why it is presented in Biogeosciences and where the other data from the GEOVIDE cruise is (or will be) presented.

The discussion section can be improved. Where increases and decreases are discussed (for example the second paragraph) the authors should mention whether this increase is statistically significant or fall with the observed variability. In the discussion on the origin of the cooling of the SPG, advection versus surface fluxes, it is important to consider the time scale that both are acting on. The changes in advection are thought to act on longer (decadal) time scales while the surface forcing has a more direct effect. In fact, the warming trend in the western SPG was halted much earlier than 2014 and much has been explained by surface forcing (Piron, 2015; de Jong and de Steur, 2016; Yashayaev and Loder, 2017). I would encourage the authors to put the 2014 anomalies into context of the recent interannual variability rather than focusing on a comparison with the mean.

Comments

- Suggest to replace MOC with AMOC (Atlantic MOC) throughout the paper since it is more appropriate in the context of the North Atlantic circulation discussed here.

- Line 63: "the ocean has taken up 90% of the heat accumulated".

- Line 74: not sure what is meant by durable.

- Lines 139-141: this criterion seems to lead to unexpected values near the shelf of Greenland. The waters in the IC and EGC are very stratified, but the orange line shows WMLD as deep as in the central Irminger Sea.

- Line 221: This current system is not commonly known as the WBC. Elsewhere the

authors refer to the East Greenland (Coastal) Current and the Irminger Current which is more appropriate.

- Line 252: "Note that the net transport in the northern branch is null".

- Line 274: "as well as it can" is not very readable. Even though I understand that the author is not a native English speaker I think the readability would improve if the authors took another critical look at the grammar of some of the sentences in this manuscript.

- Line 297: the section just seems to miss (cut south off) the high energy signal of the Irminger Current on the RR.

- Regarding the paragraphs between lines 284 and 325. The red squares in Figure 6 are mention, but the others are not. It would be easier to follow the eddy description if the yellow, green and orange squares were also denoted here.

- Lines 359-361: not clear which anomalies are referred to here. If it is the cooling/freshening in the western SPG (caused by ventilation) it is not surprising to see it linked to an oxygen increase.

- Lines 371-380: Please let the reader know where the data from these other measurements will be presented if not here.

- Line 430: Briefly mention why is 1997 excluded.

- In the discussion on freshwater surface fluxes it would be good to mention something about the uncertainties of these fields over the ocean.

---

## Referee Comment (RC3) · Anonymous Referee #3 · 21 Aug 2017

In this manuscript the authors present the results from analysis of a 2014 CTD section taken along the OVIDE section. The cruise was a contribution to the GEOTRACES program and so the results of this hydrographic analysis will allow analysis of the extensive chemistry data also collected on the cruise. The authors describe the OVIDE section from the cruise data, place it in context of data along the OVIDE section from 2002 to 2012, and attempt to explain some of the differences.

The paper is a good, straightforward description of a cruise data set, and gives a highly valuable look at the MOC and gyre circulation of the eastern subpolar North Atlantic in summer 2014. The text is well written and the figures are relevant and mainly well

presented.

The weakness of the paper lies in the authors attempt to understand the reasons for the cooling and freshening they observe in the 2014 section eastern basins compared to the earlier data. The introduction is a little muddled on this topic, and the conclusions that they draw from their analysis are not as robust as they could be. Below are some comments that I hope the authors can use to improve that aspect of the paper, as well as some minor edits.

1. Timescales of change. The authors have missed a key aspect of the literature on changes in the subpolar North Atlantic, and that is about timescales. If you could re-write the introduction considering the timescales of each of the papers that you cite it would help you focus your own analysis. In short, the arguments for ocean heat transport convergence being the primary control are all on long timescales - multiyear at least, certainly decadal. The "cold blob" analyses and the evidence for air-sea fluxes are all about short timescales - seasonal to a year or two. You should also consider the possibility that temperature anomalies and salinity anomalies may have some different forcing mechanisms on different timescales. You may not have enough data to look at long-term changes, and a focus on the short term may be more appropriate with the analysis that you have done already.

2. Methods. I realize that the authors are using well-developed methods described in earlier papers from the group, but a little bit more information would help the reader understand their method. In particular I was not sure whether the SADCP data are used in the inversion. I had thought they were, so it is surely not a surprise that the main features in the SADCP data are also seen in the solution?

I felt there should be more information about the reduced resolution of the CTD spacing in 2014. You say in lines 123-124 that you will later show that the features were "cor-rectly sampled", but your evidence for this in lines 386-387 seems to be just that all the circulation features are identified (Table 1). I would like to see this explored more - is

the higher uncertainty in Table 1 because of the resolution? If you subsample an earlier cruise at the 2014 resolution do you get the same results as the original resolution?

It would be useful to have more explanation about how you computed the gyre and overturning heat transport (lines 267-270)

3. I struggled to see the importance or relevance of section 3.2 (fronts and eddies). This looks like a description that will be useful for colleagues who are writing papers on the GEOTRACES data, but it seems to sit a little uneasily in the context of the rest of the paper. The same can be said for section 3.4. I can imagine that these lines of text could be usefully transferred into a companion paper.

4. Thermohaline anomalies. You need to state how you computed the anomalies - presumably on pressure surfaces. Your description would be more easily followed if you related the anomaly patches to the circulation features that you have already described. For example, is the first anomaly (lines 333-334) in the Irminger Current? If the anomalies are focussed in the main currents (IC, NAC) could that be evidence for ocean transport as a source of the anomalies?

An important point: I do not agree that the bottom of the anomalies is at the depth of the winter mixed layer - in most cases they extend deeper than the WMLD, which is surely significant and counter evidence for your hypothesis of air-sea fluxes being the key driver.

line 360 and elsewhere - it is best to avoid subjective words like "remarkably" especially when you do not explain what is remarkable about that observation.

5. Discussion. This section needs some improvements because the writing becomes less clear and sometimes less focussed.

Paragraph 2 (lines 392 onwards) is very unclear. I'm not always sure which data set or feature you are referring to when you quantify the transport, and how that relates to Table 1. You conclude that the Irminger Current is significantly strengthened in 2014,

but from the numbers in Table 1 it looks as they are not signficantly different within the error bars (the uncertainty on the 2014 estimates are large).

para 4 (line 413 onwards). It is interesting to me that the SAF has shifted southeastward (by how much?). Does this actually imply that the Bersch mechanism for freshening of the eastern basin might be at work, even though you are arguing for this not being the source of the freshening? You need to explain why the Marsh and Grist papers are relevant to this work, since they refer to a different branch of the NAC that does not come here - what is the connection?

Finally, I come back to my point about timescales of forcing. I think your result that the heat transport is high even though the upper ocean is cooler is interesting, but I don't agree that it is necessarily contrasting with the results of Desbruyeres et al. The air-sea fluxes that you present are a great result - but they are only for 1 year, and many of the papers that you refer to are talking about ocean transport convergence as the primary factor over longer timescales. It is not correct to say that the 2014 anomaly comes after 18 years of warming and salinification - papers by Robson, Holliday, and the ICES Report on Ocean Climate show observed declining temperatures and salinity in these eastern basins since the late 2000s, part of multi-year variability. That said, the obs also show a sharp drop in salinity and temperature in recent years, so it seems likely that the longer term changes in circulation are being reinforced by enhance air-sea fluxes. It might help if you focused the discussion on a short term atmospheric influence, superimposed on a longer term trend.

Minor edits

- Lines 88-89, Holliday et al does not look at the Irminger Sea. Their hypothesis was about a shift of the subpolar front (the Bersch mechanism) not advection.

- line 144 (and elsewhere) "hydrological" should be replaced with "hydrographic"

- line 219 use "Fig. 3b" rather than "Fig. 3, lower panel"

- line 239 what do you mean by barotropic streamfunction here? You are referring I think to the plot of accumulated transport - is that the same thing?

- line 246 and Fig. 5, I find the green dots hard to see - can you use a color that stands out more clearly?

———————————————————

---

## Author Comment (AC1) · 2 Oct 2017

Dear editor and referees,

We were grateful to receive the very constructive reviews of our paper "The GEOVIDE cruise in May-June 2014 reveals an intense Meridional Overturning Circulation over a cold and fresh subpolar North Atlantic". Thank you very much to the 3 anonymous referees. We will incorporate in the manuscript the majority of their comments and we think the scientific results will be better exposed than in the first submission. Before dealing with the referee comments in detail, we wrote an answer to a concern that is common to the three reviews: the misunderstanding about the timescales dominating

the cooling and freshening of the subpolar North Atlantic. Following, we answer point by point each comment of the three referees.

One result of our paper is the co-existence in May – June 2014 of the cooler and fresher eastern subpolar North Atlantic (SPNA) in relation to the mean 2002 – 2012, and the relative intense Meridional Overturning Circulation and heat transport across the OVIDE section. In the region delimited by 40°N – 60°N, 45°W – 10°W, the evolution of both (i) ocean heat and freshwater content in the upper 1000 m and (ii) air-sea fluxes of heat and freshwater since 2013 reveals that the atmospheric forcing is mostly responsible for the strong TS anomalies of 2014. However, as pointed by the three reviewers, we did not discuss the decadal context of our observations and missed to refer to Robson et al. (2016; 2017) who identified a new cooling period in the subpolar North Atlantic starting in mid-2000, a cooling period affecting at decadal time-scale.

In our revision, we will include a new figure showing the evolution of heat content in the upper 1000m of the region 40°N – 60°N, 45°W – 10°W (see Figure 1 in this document). Based on this figure, that shows a long term heat content decrease starting in the mid-2000s, we will be able to illustrate Robson et al. (2016; 2017). We also observe intensification in heat content decrease from 2013 to 2014, just the episode we discuss in our paper. Thus, the 2013-2014 cooling episode is inserted in the cooling at longer period of time detected by Robson et al. (2016; 2017). We show in our paper that the former was dominantly produced by the local atmospheric forcing over the 2013-2014 winter. Our results are in agreement with a recent paper, Frajka-Williams et al., (2017, in Scientific Reports): they exposed that the rapid cooling registered between 2013 and 2015 in the subpolar North Atlantic was explained by the atmospheric forcing since the effects of a MOC slow-down at 26°N is too slow to explain the observed rapid cooling.

Following this explanation, the third paragraph of the introduction will be restructured and the references to Robson et al. (2016-2017) added. The discussion 4.2 will also be expanded in order to place our observations in the context of a longer time scale. The figure 1 in this document will be inserted in the new ms.
We copied the referee comments in this document in blue font followed by our answers in black font.

Anonymous Referee 2

The GEOVIDE cruise in May–June 2014 reveals an intense Meridional Overturning Circulation over a cold and fresh subpolar North Atlantic, by Patricia Zunino et al. This paper discusses the physical background of the GEOVIDE cruise in 2014. It highlights changes in transport as well as heat and freshwater content compared to the 2002-2012 mean state. The most interesting conclusion is that the large scale cooling seen in the SPG is (more then) compensated by a strengthened circulation in the net heat transport.

One comment on the structure of the paper. Although the TEI measurements do not feature in the abstract there are discussed several times in the introduction and elsewhere in the paper. It therefore reads like the discussion of the TEI measurements will be discussed later in the paper, but this never happens. It is also never mentioned where (in which other paper) these measurements will be shown. This does not improve the overall clarity of the paper. The authors should rephrase the text as to make it more clear what the focus of this paper is, why it is presented in Biogeosciences and where the other data from the GEOVIDE cruise is (or will be) presented.

We understand that it can be somehow uncommon to publish a physical paper in Biogeoscience. However, our paper is part of a special issue in Biogeosciences with all the papers resulting from the GEOVIDE cruise. This is why our paper, which defines the physical background of the GEOVIDE cruise, has been submitted to Biogeoscience. In order to make it clearer, we will introduce some references at the end of the introduction and at the beginning of the section 3.4 in the new version of the manuscript.

The discussion section can be improved. Where increases and decreases are discussed (for example the second paragraph) the authors should mention whether this increase is statistically significant or fall with the observed variability.

Ok, we think that we did not introduce enough the ideas in each paragraph, so we will always begin the paragraph with a description of the transport anomaly, including its significance. Actually, we only discussed the NAC and Irminger C. that showed a significant variability in their different components, although their overall transport did not. Paragraph 2 of the discussion (line 392) could begin by: "When defining the IC as in D2016, we saw an increase in the IC intensity in 2014, but within the observed variability (table 1). However, the such-defined IC encompasses a warm and salty northward transport and a cold and fresh southward transport. So, to go further [. . .]".

At the beginning of paragraph 3 (line 402), we propose: "Concerning the weaker NAC, its 2014 intensity, 32.2 $\pm$ 11.4 Sv, is weaker although in the limits of the observed variability (41.8 $\pm$ 3.7 Sv) in 2014. By the decomposition of this wide current, it is very likely [. . .]"

In the discussion on the origin of the cooling of the SPG, advection versus surface fluxes, it is important to consider the time scale that both are acting on. The changes in advection are thought to act on longer (decadal) time scales while the surface forcing has a more direct effect. In fact, the warming trend in the western SPG was halted much earlier than 2014 and much has been explained by surface forcing (Piron, 2015; de Jong and de Steur, 2016; Yashayaev and Loder, 2017). I would encourage the authors to put the 2014 anomalies into context of the recent interannual variability rather than focusing on a comparison with the mean.

This is the general critique done by the three reviewers. We agree that the changes in the lateral advection affect the heat and freshwater content changes at decadal timescale. We also agree that the air-sea flux is thought to cause heat and freshwater changes in the ocean at shorter period of time, buffering or intensifying the effect of the lateral advection. A more detailed answer is given in the introduction of this document.

We will reorganize and expand the third paragraph introduction and discussion of the ms in order to be clearer in the timescales of both processes creating thermohaline anomalies in the SPNA (see also answers to reviewer 1 comments).

Comments

- Suggest to replace MOC with AMOC (Atlantic MOC) throughout the paper since it is more appropriate in the context of the North Atlantic circulation discussed here.

We agree that the MOC is a general term for the Meridional Overturning Circulation affecting all the oceans. So we will use "AMOC" in the introduction, but keep "MOC" in the result for consistency with other references (Mercier et al. 2015 and Daniault et al. 2016). Actually, we should even call it OVIDE MOC in the results but we prefer to keep it simple.

- Line 63: "the ocean has taken up 90% of the heat accumulated".

Right, we will modify it.

- Line 74: not sure what is meant by durable.

It means persistent in time. In any case we will rewrite that sentence as: "Robson et al. (2012) found that the rapid warming of the SPNA was primarily caused by a relative long period of high northward ocean heat transport associated with the strengthening (. . .)".

- Lines 139-141: this criterion seems to lead to unexpected values near the shelf of Greenland. The waters in the IC and EGC are very stratified, but the orange line shows WMLD as deep as in the central Irminger Sea.

It was a mistake, there should have been no value here, Actually, the WMLD cannot be determined west of station 48 because of the strong layering in the East Greenland/Irminger Current.

- Line 221: This current system is not commonly known as the WBC. Elsewhere the

authors refer to the East Greenland (Coastal) Current and the Irminger Current which is more appropriate.

The "western boundary current" is used in many papers referring to the currents over whole water column in the western side of the oceans (see for exemple "Moored Observations of Western Boundary Current Variability and Thermohaline Circulation at 26.5° in the Subtropical North Atlantic" of Lee et al. (1996) or "The western boundary current of the seasonal subtropical gyre in the Bay of Bengal" of Shetye et al. (1993). In our region, we have referred to this dynamic structure as WBC following Daniault et al. (2016). The Western Boundary Current was defined as the sum of the East Greenland Irminger Current and the Deep Western Boundary Current. The former is composed by three components: the East Greenland Current, the spill jet and the adjacent IC that usually cannot be separated at 60°N. In our paper we were referring to the water flowing southward at the western side of the Irminger Sea, west and below the Irminger Gyre. Furthermore, when we talked about the Irminger Current, we were specifically referring to the water flowing northeastward west of the Reykjanes Ridge, although it is true that it recirculates in the East Greenland/Irminger Current (Pickart et al., 2005). In any case, we will rewrite this part of the manuscript to be clearer, but we consider that WBC is the appropriate term here.

- Line 252: "Note that the net transport in the northern branch is null".

Thank you, we will simplify this sentence.

- Line 274: "as well as it can" is not very readable. Even though I understand that the author is not a native English speaker I think the readability would improve if the authors took another critical look at the grammar of some of the sentences in this manuscript.

Ok, we will change "as well as it can . . ." by "and they can . . . ". Yes, we are not English speaker, this is why our ms was already revised by a professional native English speaker.

- Line 297: the section just seems to miss (cut south off) the high energy signal of the Irminger Current on the RR.

We do not understand what the referee refers to. In this paragraph, we are discussing about the large eddy near Eriador Seamount. The Irminger Current is quite far away to the northwest.

- Regarding the paragraphs between lines 284 and 325. The red squares in Figure 6 are mention, but the others are not. It would be easier to follow the eddy description if the yellow, green and orange squares were also denoted here.

Totally right. Actually, the colored squares were introduced in the figure to make easier the eddy identifications. In the new version of the manuscript we will better indicate all of them.

- Lines 359-361: not clear which anomalies are referred to here. If it is the cooling/freshening in the western SPG (caused by ventilation) it is not surprising to see it linked to an oxygen increase.

Maybe the term "zooming out" was misunderstood. When we talked about ventilation, we were indeed referring to the oxygen positive anomaly, so we will precise Figure 7c. This paragraph was written to give a general view of the link between temperature, salinity and oxygen anomalies. So we will slightly modify this sentence to make it clearer.

- Lines 371-380: Please let the reader know where the data from these other measurements will be presented if not here.

We will include references to the other papers of the Geovide special issue in Biogeoscience in the introduction and in section 3.4.

- Line 430: Briefly mention why is 1997 excluded.

We modified the sentence as "To compare it with the 2002-2010 average, we used the

data of Mercier et al. (2015), without data from 1997 because it did not belong to our reference period, [. . .]".

- In the discussion on freshwater surface fluxes it would be good to mention something about the uncertainties of these fields over the ocean.

Right, we already thought about this complex issue, because uncertainties in evaporation and precipitation products are very difficult to assess (Dee et al., 2011). So Josey and Marsh (2005) made an estimate by comparing NCEP and ERA-40 products. We did the same in our case. The difference between monthly freshwater fluxes over the region 40°N-60°N, 45°W-10°W estimated with NCEP and ERA-INTERIM is about 10% of the absolute values. Over 2002-2015, no clear bias stands out between both products and accumulating fresh water fluxes over years does not increase the relative difference of 10%. The accumulated air-sea freshwater flux of 2 x $10^{12}$ $m^3$ from 2013 to 2014 that was shown in Figure 10 (ERA-IN) is different by 2 x $10^{11}$ $m^3$ from the NCEP estimate; we will discuss about these errors for the period 2013 to 2014 in the new version of the ms and add the NCEP estimate in Figure 10.
* * *
[Figure]

Figure 1: Heat content anomalies in relation to the mean heat content for the period 2002 - 2012 in the upper 1000m of the region 40°N-60°N and 45°W-10°W. Grey line is the monthly time series; black line is the 2-year running mean of the monthly time series. Data source: EN4 database (Good et al. 2013).

**Fig. 1.**

[Figure]

Figure 2: Contours of the Absolute Dynamical Topography averaged over 2014 (in thin lines). Contours are every 0.05m. Thick contours correspond to the levels encompassing the SAF front during OVIDE cruises: bold red lines for the mean 2002 – 2012 and bold black lines for 2014. Note that the temporal trend on the mean ADT over the whole box (2.8mm/yr) was removed. Bathymetry (1000m step contours) and the OVIDE section are plotted in white. Colors represent the absolute velocity of the current (yellow for velocities stronger than 0.3m/s). This figure will be added to the new ms.

**Fig. 2.**

---

## Author Comment (AC2) · 2 Oct 2017

Dear editor and referees,

We were grateful to receive the very constructive reviews of our paper "The GEOVIDE cruise in May-June 2014 reveals an intense Meridional Overturning Circulation over a cold and fresh subpolar North Atlantic". Thank you very much to the 3 anonymous referees. We will incorporate in the manuscript the majority of their comments and we think the scientific results will be better exposed than in the first submission. Before dealing with the referee comments in detail, we wrote an answer to a concern that is common to the three reviews: the misunderstanding about the timescales dominating

the cooling and freshening of the subpolar North Atlantic. Following, we answer point by point each comment of the three referees.

One result of our paper is the co-existence in May – June 2014 of the cooler and fresher eastern subpolar North Atlantic (SPNA) in relation to the mean 2002 – 2012, and the relative intense Meridional Overturning Circulation and heat transport across the OVIDE section. In the region delimited by 40°N – 60°N, 45°W – 10°W, the evolution of both (i) ocean heat and freshwater content in the upper 1000 m and (ii) air-sea fluxes of heat and freshwater since 2013 reveals that the atmospheric forcing is mostly responsible for the strong TS anomalies of 2014. However, as pointed by the three reviewers, we did not discuss the decadal context of our observations and missed to refer to Robson et al. (2016; 2017) who identified a new cooling period in the subpolar North Atlantic starting in mid-2000, a cooling period affecting at decadal time-scale.

In our revision, we will include a new figure showing the evolution of heat content in the upper 1000m of the region 40°N – 60°N, 45°W – 10°W (see Figure 1 in this document). Based on this figure, that shows a long term heat content decrease starting in the mid-2000s, we will be able to illustrate Robson et al. (2016; 2017). We also observe intensification in heat content decrease from 2013 to 2014, just the episode we discuss in our paper. Thus, the 2013-2014 cooling episode is inserted in the cooling at longer period of time detected by Robson et al. (2016; 2017). We show in our paper that the former was dominantly produced by the local atmospheric forcing over the 2013-2014 winter. Our results are in agreement with a recent paper, Frajka-Williams et al., (2017, in Scientific Reports): they exposed that the rapid cooling registered between 2013 and 2015 in the subpolar North Atlantic was explained by the atmospheric forcing since the effects of a MOC slow-down at 26°N is too slow to explain the observed rapid cooling.

Following this explanation, the third paragraph of the introduction will be restructured and the references to Robson et al. (2016-2017) added. The discussion 4.2 will also be expanded in order to place our observations in the context of a longer time scale. The figure 1 in this document will be inserted in the new ms.

We copied the referee comments in this document in blue font followed by our answers in black font.

Anonymous Referee 3

In this manuscript the authors present the results from analysis of a 2014 CTD section taken along the OVIDE section. The cruise was a contribution to the GEOTRACES program and so the results of this hydrographic analysis will allow analysis of the extensive chemistry data also collected on the cruise. The authors describe the OVIDE section from the cruise data, place it in context of data along the OVIDE section from 2002 to 2012, and attempt to explain some of the differences.

The paper is a good, straightforward description of a cruise data set, and gives a highly valuable look at the MOC and gyre circulation of the eastern subpolar North Atlantic in summer 2014. The text is well written and the figures are relevant and mainly well presented. The weakness of the paper lies in the authors attempt to understand the reasons for the cooling and freshening they observe in the 2014 section eastern basins compared to the earlier data. The introduction is a little muddled on this topic, and the conclusions that they draw from their analysis are not as robust as they could be. Below are some comments that I hope the authors can use to improve that aspect of the paper, as well as some minor edits.

1. Timescales of change. The authors have missed a key aspect of the literature on changes in the subpolar North Atlantic, and that is about timescales. If you could rewrite the introduction considering the timescales of each of the papers that you cite it would help you focus your own analysis. In short, the arguments for ocean heat transport convergence being the primary control are all on long timescales – multiyear at least, certainly decadal. The "cold blob" analyses and the evidence for air-sea fluxes are all about short timescales - seasonal to a year or two. You should also consider the possibility that temperature anomalies and salinity anomalies may have some different
forcing mechanisms on different timescales. You may not have enough data to look at long-term changes, and a focus on the short term may be more appropriate with the analysis that you have done already.

Yes, as indicated in the introduction of this document, we agree that the mechanisms controlling the heat and freshwater content changes in the ocean at different time-scales were not well exposed in the manuscript. We will reorganize the last part of the introduction and the last part of the discussion to account for this. We will put the "inter-annual" signal of 2013-2014, which is an intensification of the cooling already started in mid-2000s as documented by Robson et al. (2016;2017), and shown in this manuscript to be dominantly caused by the air-sea flux.

2. Methods. I realize that the authors are using well-developed methods described in earlier papers from the group, but a little bit more information would help the reader understand their method. In particular I was not sure whether the SADCP data are used in the inversion. I had thought they were, so it is surely not a surprise that the main features in the SADCP data are also seen in the solution?

Yes, as it is indicated in the manuscript in line 144 – 145, the SADCP data are used to constrain the inverse model. To make it clearer, just after the first sentence, we will add the sentence "Before inversion, the S-ADCP data were averaged between stations in layers where the velocity vertical shear was consistent with the geostrophic velocity profiles". The referee is right: it is not a surprise that the main features in the SADCP data are also visible in the inversion solution. However, we wanted to show how the small-scale features visible in the SADCP data were averaged in the inversion because of the coarser resolution of the hydrographic profiles (lines 161-163).

I felt there should be more information about the reduced resolution of the CTD spacing in 2014. You say in lines 123-124 that you will later show that the features were "correctly sampled", but your evidence for this in lines 386-387 seems to be just that all the circulation features are identified (Table 1). I would like to see this explored more – is

the higher uncertainty in Table 1 because of the resolution? If you subsample an earlier cruise at the 2014 resolution do you get the same results as the original resolution?

There are several points in your questions. First, we did sensitivity study on the 2010 data to determine the optimized sampling for the GEOVIDE cruise. The 2010 transport data showed the same results with the original high resolution and the GEOVIDE resolution, but indeed, errors on regional features increased when subsampled. However, in 2014, we used an OS38 (the latest generation of SADCP with a 1200m range), and could average the constrains in a deeper layer (more geostrophic) with less uncertainties that all the previous surveys where a NB75 was used. This is why, at the end, the errors in transports are quite similar in GEOVIDE and in the previous OVIDE surveys. This information will be synthetized in section 2.2 in the new version of the manuscript. Second, in table 1, the 2014 errors were calculated from the covariance matrix resulting from the box inverse model, while the errors given for the means (2002 - 2012) were standard deviations of the six estimates of the different currents. In fact, because the transport estimates of the currents are more or less stable during the 2002 – 2012 period, their standard deviations are low compared to the error given for each single OVIDE cruise. The information about how the errors were computed will be introduced in the table caption in the new version of the manuscript.

It would be useful to have more explanation about how you computed the gyre and overturning heat transport (lines 267-270)

We will add "(their equation (1))" line 269 after referring to Mercier et al. (2015). We prefer not to expand this topic because it is not a new result: one of the most important results of Mercier et al. was indeed that the MOC was the primary driver of the heat flux across the OVIDE section. We just wanted to complement their time series.

3. I struggled to see the importance or relevance of section 3.2 (fronts and eddies). This looks like a description that will be useful for colleagues who are writing papers on the GEOTRACES data, but it seems to sit a little uneasily in the context of the rest

of the paper. The same can be said for section 3.4. I can imagine that these lines of text could be usefully transferred into a companion paper.

Yes, we agree that 3.2 and 3.4 sections are not so relevant as results. Nevertheless, one of the objectives of this paper was to define the physical background of the GEO-VIDE cruise, which is very important for the interpretation of the TEIs distribution to be carried out by our GEOTRACES colleagues. In fact, this paper has been submitted to Biogeoscience Discussion as part of the GEOVIDE Special Issue, where all the GEOVIDE papers are going to be submitted. So, we consider that we can leave both sections in our physical paper, inside this Biogeoscience Special Issue. Nevertheless, we will introduce more information about the other GEOVIDE papers in section 3.4 in the new version of the manuscript.

4. Thermohaline anomalies. You need to state how you computed the anomalies - presumably on pressure surfaces.

It was indicated how the anomalies were estimated at the beginning of the first paragraph of section 3.3 as:" In the following, S and T anomalies were quantified as the mean values of the anomaly patches represented in Fig. 7." Following the remark of the reviewer, we will add in this paragraph that those anomalies were computed in pressure coordinates. Density coordinates are generally more appropriate, but it makes the interpretation much trickier and does not substantially change the conclusions of our study.

Your description would be more easily followed if you related the anomaly patches to the circulation features that you have already described. For example, is the first anomaly (lines 333-334) in the Irminger Current?

Good idea. So we will transform the sentence as: "First, negative anomalies in surface waters were observed over the RR (in the IC and the ERRC), and east of 20°W (in the SNAC and its recirculation)". The deeper anomalies are associated with the variability of waters masses (LSW and ISOW) and not specifically to dynamical features. The

anomalies in the Mediterranean Water were already associated with eddies.

If the anomalies are focussed in the main currents (IC, NAC) could that be evidence for ocean transport as a source of the anomalies?

Lines 355-358 suggest that the displacement of the SAF (i.e. central NAC) is preponderant in the fresh and cold anomalies at 23°W. However, we did not really interpret the anomalies in the ERRC and IC with the lateral advection, although, as you suggest, it is surely an advected signal. See below for a more precise answer.

An important point: I do not agree that the bottom of the anomalies is at the depth of the winter mixed layer - in most cases they extend deeper than the WMLD, which is surely significant and counter evidence for your hypothesis of air-sea fluxes being the key driver.

We agree, we did not put enough weight on the advective origin of some anomalies, and precisely the anomaly in the ERRC. Thank you for this interesting remark. So, to answer your remark and improve the manuscript, we will reformulate lines 333-336 by: "First, negative anomalies in surface-intermediate waters were observed above the WMLD over the Reykjanes Ridge (IC and ERRC) and east of 20° W (in the SNAC and its recirculation). In the former, the S and theta anomalies were quantified at -0.07 and -0.95°C, respectively. In the latter, the negative anomalies of S and theta amounted to -0.11 and -0.70°C. In the ERRC, negative S and theta anomalies also appeared below the WMLD, but concerns a water mass that is different from the one in the WMLD (Fig 2b); both water masses are separated by a maximum of potential vorticity (not shown)."
Then, in the discussion, we will also reformulate lines 484-487: More evidence for the important role of air-sea fluxes is provided by the distribution of theta-S and oxygen anomalies in the water column. Indeed, the WMLD along the OVIDE section east of the 20 °W coincided with the deep limit of the anomalies (Fig. 7). In the ERRC, the WMLD crosses the anomaly separating SPMW and upper LSW (Fig. 2b), both being advected together by the ERRC, but probably issued from different ventilation regions.

[Figure]

According to Boisséson et al. (2012), the SPMW is formed by air-sea interactions on its way around the Iceland basin."

line 360 and elsewhere - it is best to avoid subjective words like "remarkably" especially when you do not explain what is remarkable about that observation.

Ok, we checked the whole ms and we can and we will remove this adverb everywhere, and reformulate when necessary.

5. Discussion. This section needs some improvements because the writing becomes less clear and sometimes less focussed. Paragraph 2 (lines 392 onwards) is very unclear. I'm not always sure which data set or feature you are referring to when you quantify the transport, and how that relates to Table 1. You conclude that the Irminger Current is significantly strengthened in 2014, but from the numbers in Table 1 it looks as they are not significantly different within the error bars (the uncertainty on the 2014 estimates are large).

We will make the following changes to improve the clarity of the message. First, we will introduce the paragraph 2 by :"When defining the IC as in D2016, we saw an increase in the IC intensity in 2014, but within the observed variability (table 1). However, the such-defined IC encompasses a warm and salty northward transport and a cold and fresh southward transport. So, to go further [. . .]". We will also add a specific column in Table 1 for the transport of the part of the IC that flows northward, which is the one that differs significantly from the mean of D2016.

para 4 (line 413 onwards). It is interesting to me that the SAF has shifted southeastward (by how much?).

It is indicated in the manuscript, approximately 100 km along the OVIDE section (line 414), and this is consistent with the SAF displacement observed in the ADT (see Figures 2 in this document and comments in the answer to Reviewer 1). To make it clearer, we will add to the ms the ADT figure (Figure 2 in this document).

*Does this actually imply that the Bersch mechanism for freshening of the eastern basin might be at work, even though you are arguing for this not being the source of the freshening?*

At the end of the discussion of the manuscript we briefly discussed about the Bersch mechanism. Inspired by your remark, we pushed further and answer no, our observations lead us to conclude that the freshening at short timescales is not associated with more advection of subpolar water into the eastern SPNA. So we will remove the last paragraph of the discussion (lines 520-527) that was related to the observations of Bersch et al., but add this information to the paragraph about the southeastward displacement of the SAF (line 413-418), as follows: "The SAF, that bears the central branch of the NAC, shows also a remarkable southeastward displacement in 2014 in relation to the mean circulation pattern (station 26 in Fig. 1), of about 100 km. A careful study of the ADT streamlines showed that this displacement was not due to a peculiar meandering of the front (Fig 8). Bersch et al. (2007) linked the displacement of the SAF with the NAO. After a decade of neutral winter NAO index, it turned positive in 2011 and continued positive in 2013 and very high in 2014 (Hurrell et al., 2017). Therefore, symmetrically with Bersch et al. (2007), the southeastwards displacement of the SAF is consistent with their observations. However, following the mechanisms proposed by Bersch et al. (2007), an enhanced eastward advection of cold and fresh water in the eastern SPNA would also be expected, and conversely, we found an increase in the heat transport across the section even if we also found cold and fresh anomalies. This observation will be further discussed later in the light of air-sea fluxes".

*You need to explain why the Marsh and Grist papers are relevant to this work, since they refer to a different branch of the NAC that does not come here - what is the connection?*

Yes, the referee is right, they referred to a different branch of the NAC. The connection here is just to compare our result about the southeastward displacement of the SAF with other works, but obviously, it blurs the picture since it is further south, so we will

delete these 2 sentences about Grist et al. . .

Finally, I come back to my point about timescales of forcing. I think your result that the heat transport is high even though the upper ocean is cooler is interesting, but I don't agree that it is necessarily contrasting with the results of Desbruyeres et al. The air-sea fluxes that you present are a great result - but they are only for 1 year, and many of the papers that you refer to are talking about ocean transport convergence as the primary factor over longer timescales.

Yes, we agree with the referee, the explanation of the changes at different time scales and the mechanisms generating them is the first point to improve in this manuscript. In relation to the Desbruyères et al., we want to keep the contrast with Desbruyères et al. (2015), making it clearer, so we propose in line 449-453 this sentence instead: "This result might be the effect of a short term variability since it contrasts with the study of Desbruyères et al. (2015), who argued that the long-term variability of the ocean heat transport at the OVIDE section is dominated by the advection by the mean velocity field of temperature anomalies formed upstream rather than the velocity anomalies acting on temperature".

It is not correct to say that the 2014 anomaly comes after 18 years of warming and salinification - papers by Robson, Holliday, and the ICES Report on Ocean Climate show observed declining temperatures and salinity in these eastern basins since the late 2000s, part of multi-year variability.

Yes, totally right, the sentence "The 2014 anomaly was the first detected, after approximately 18 years of warming and salinification" is going to be deleted in the new version of the manuscript because it is wrong.

That said, the obs also show a sharp drop in salinity and temperature in recent years, so it seems likely that the longer term changes in circulation are being reinforced by enhance air-sea fluxes. It might help if you focused the discussion on a short term atmospheric influence, superimposed on a longer term trend.

Exactly, as it has been previously exposed in this document, this is what we are going to do in the new version of the manuscript. We will even add "at short time scale" in the last sentence of the abstract: "We concluded that, at short time scale, these changes were mainly driven by air-sea heat and freshwater fluxes rather than by ocean circulation".

Minor edits

- Lines 88-89, Holliday et al does not look at the Irminger Sea. Their hypothesis was about a shift of the subpolar front (the Bersch mechanism) not advection.

Right only Iceland basin, we will remove Irminger in this line. About their hypothesis, actually, we were referring to the sentence "The decrease in potential temperature and salinity after 2010 in all basins provides the first new evidence that the eastern subpolar Atlantic is once again influenced by cold, fresh western subpolar water." (paragraph 5, section 5 of Holliday et al. (2015)". So we will explicitly quote Holliday et al. to clarify.

- line 144 (and elsewhere) "hydrological" should be replaced with "hydrographic"

Yes, we will change it everywhere.

- line 219 use "Fig. 3b" rather than "Fig. 3, lower panel"

Yes, right.

- line 239 what do you mean by barotropic streamfunction here? You are referring I think to the plot of accumulated transport - is that the same thing?

Yes, the barotropic streamfunction is the volume transport vertically accumulated and horizontally accumulated from Greenland to each station. We will indicate it in the new version of the manuscript.

- line 246 and Fig. 5, I find the green dots hard to see - can you use a color that stands out more clearly? Yes, we will change it to white dots.

[Figure]

[Figure]

Figure 1: Heat content anomalies in relation to the mean heat content for the period 2002 - 2012 in the upper 1000m of the region 40°N-60°N and 45°W-10°W. Grey line is the monthly time series; black line is the 2-year running mean of the monthly time series. Data source: EN4 database (Good et al. 2013).

**Fig. 1.**

[Figure]

Figure 2: Contours of the Absolute Dynamical Topography averaged over 2014 (in thin lines). Contours are every 0.05m. Thick contours correspond to the levels encompassing the SAF front during OVIDE cruises: bold red lines for the mean 2002 – 2012 and bold black lines for 2014. Note that the temporal trend on the mean ADT over the whole box (2.8mm/yr) was removed. Bathymetry (1000m step contours) and the OVIDE section are plotted in white. Colors represent the absolute velocity of the current (yellow for velocities stronger than 0.3m/s). This figure will be added to the new ms.

**Fig. 2.**

---

## Author Comment (AC3) · 2 Oct 2017

Dear editor and referees,

We were grateful to receive the very constructive reviews of our paper "The GEOVIDE cruise in May-June 2014 reveals an intense Meridional Overturning Circulation over a cold and fresh subpolar North Atlantic". Thank you very much to the 3 anonymous referees. We will incorporate in the manuscript the majority of their comments and we think the scientific results will be better exposed than in the first submission. Before dealing with the referee comments in detail, we wrote an answer to a concern that is common to the three reviews: the misunderstanding about the timescales dominating

the cooling and freshening of the subpolar North Atlantic. Following, we answer point by point each comment of the three referees.

One result of our paper is the co-existence in May – June 2014 of the cooler and fresher eastern subpolar North Atlantic (SPNA) in relation to the mean 2002 – 2012, and the relative intense Meridional Overturning Circulation and heat transport across the OVIDE section. In the region delimited by $40°N – 60°N, 45°W – 10°W$, the evolution of both (i) ocean heat and freshwater content in the upper 1000 m and (ii) air-sea fluxes of heat and freshwater since 2013 reveals that the atmospheric forcing is mostly responsible for the strong TS anomalies of 2014. However, as pointed by the three reviewers, we did not discuss the decadal context of our observations and missed to refer to Robson et al. (2016; 2017) who identified a new cooling period in the subpolar North Atlantic starting in mid-2000, a cooling period affecting at decadal time-scale.

In our revision, we will include a new figure showing the evolution of heat content in the upper 1000m of the region $40°N – 60°N, 45°W – 10°W$ (see Figure 1 in this document). Based on this figure, that shows a long term heat content decrease starting in the mid-2000s, we will be able to illustrate Robson et al. (2016; 2017). We also observe intensification in heat content decrease from 2013 to 2014, just the episode we discuss in our paper. Thus, the 2013-2014 cooling episode is inserted in the cooling at longer period of time detected by Robson et al. (2016; 2017). We show in our paper that the former was dominantly produced by the local atmospheric forcing over the 2013-2014 winter. Our results are in agreement with a recent paper, Frajka-Williams et al., (2017, in Scientific Reports): they exposed that the rapid cooling registered between 2013 and 2015 in the subpolar North Atlantic was explained by the atmospheric forcing since the effects of a MOC slow-down at $26°N$ is too slow to explain the observed rapid cooling.

Following this explanation, the third paragraph of the introduction will be restructured and the references to Robson et al. (2016-2017) added. The discussion 4.2 will also be expanded in order to place our observations in the context of a longer time scale. The figure 1 in this document will be inserted in the new ms.

We copied the referee comments in this document in blue font followed by our answers in black font.

Anonymous Referee 1

One of the main findings in this paper is that despite the ongoing cooling/freshening anomaly in the SPNA, the authors measure stronger heat transport across the OVIDE section during the GEOVIDE cruise in May-June 2014. The authors further conclude by strongly stating that air-sea (heat and freshwater) fluxes were the dominant factor for the observed changes and that ocean circulation played a minor role. The authors seem to have ignored the fact that the subpolar gyre (SPG) cooling/freshening is part of a decadal trend most likely initiated by ocean advection and circulation. There are evidence that the SPG temperature and salinity reversed already in 2005 (e.g. Robson et al. 2016, Nature Geoscience), and that the NAO shifted more to the positive state later on around 2011. The 2014 cooling/freshening thus may have been enhanced by air-sea fluxes (not locally formed), but it is important to keep in mind that the cooling as well as the freshening did not start in 2014 as the authors mention. It was well underway! Nevertheless, I think the paper is good and taking into account what I have mentioned above as well as my comments below will help improve the paper, which I recommend for publication after these issues have been resolved.

Major comments

I think the authors need to focus on the 2014 event as being part of the (multi-)decadal cooling/freshening rather than the instigator of it. A robust discussion along these lines is therefore strongly recommended.

Yes, we agree that the 2014 event was not the instigator; it appears to add to the decadal cooling started in 2005 and linked by Robson et al. (2016; 2017) to ocean circulation and heat transport. This point will be clarified in the discussion.

Furthermore, I wonder also how the authors reconcile the fact that there are numerous studies (see e.g. Robson et al. 2017, Clim Dyn, and references therein) demonstrating that the western SPG is dominated by surface fluxes, while the eastern SPG is dominated by ocean advection.

We agree with the reviewer this applies on the decadal time scales. Our study focuses on an inter-annual event dominantly forced by local air-sea fluxes that appears to add to the decadal signal.

How do you explain the large-scale salinity anomaly in Fig. 8 that spans both the SPNA and the region of the Gulf Stream? there is very little discussed about this basin-scale feature.

Your comment is very interesting and further investigations in the Gulf Stream region would be necessary to determine whether it is a coherent basin-scale feature especially because the negative salinity anomaly is associated with a positive temperature anomaly in the Gulf Stream Region but to a negative temperature anomaly in the SPNA. Our objective is to understand what was observed during the cruise in 2014. In order to interpret the large-scale salinity anomaly, we estimated the freshwater content change in the box $40°N - 60°N$, $45°W - 10°W$ using the ISAS data shown on Figure 8, data from other data sources (EN4 and JAMSTEC), as well as air-sea freshwater flux (ERA-interim and NCEP), see Figure 10 in the initial ms. The surface freshwater fluxes in the eastern SPNA were found to be important in the observed salinity anomaly.

Comments I think the calculations and results are straightforward, but I have some comments that can help to further improve the paper: Using the inverse model the authors and thereby identify all main flows, and show that the ship-ADCP velocities are largely similar, at least in structure, to those based on the inverse modelling. I suggest to compare your transport numbers, whenever possible, to other studies for a complete picture.

We understand your suggestion. In fact, comparing transport numbers with previous

studies was extensively done by Daniault et al. (2016), D2016 hereafter, who described the mean state of the circulation in the SPNA based on OVIDE data and the existing literature (e.g. Rossler et al. (2015), Sarafanov et al. (2008; 2012), Väge et al. (2008; 2011)). In this study, we only compare with results in other works when significant differences with D2016 were found (for example Väge et al., 2011).

Weaker NAC during 2014 is an interesting finding. The top-to-bottom transport of 0 ± 6Sv as compared to the 11 ± 4 Sv estimated by D2016 is large. I believe it is possible to show this large-scale shift from altimetry along the eddy-blockage and the doubling of the intensity of the SAF, which the authors are briefly mentioning in lines 404-412. Please elaborate also on the transfer of transport from the northern to the central branch, it is not clear to me how this occur!

We found your idea very good, and we plotted the mean ADT contours for 2002-2012 and for 2014 after removing the trend in the sea-level rise (2.8mm/yr in our region), see Fig. 2 and 3, respectively, in this document. The colors show the current velocity and highlight the energetic areas. Then we plotted the stream lines encompassing the SAF in bold; they have the same ADT values in both figures (Fig 2. and Fig. 3 in this document) and represent a slope of 15 cm in the ADT. We also plotted in both figures a red circle at the 2014 SAF position (station 26 of GEOVIDE). To interpret this figure, we assume that the bold streamlines delimit the travel of the surface waters crossing the OVIDE section at the SAF position. For comparison, we added in Fig. 3 the 2002 – 2012 mean stream lines encompassing the SAF in red. We see that along the OVIDE section, the SAF is quite narrow and located more to the southeast in 2014, when compared to the 2002-2012 mean. But more important, the NAC transport seems to be dispatched on a larger area in the mean, encompassing the permanent eddy characteristic of the northern branch of the NAC. So in the mean, the NAC transport is shared among both branches, while in 2014, it is concentrated in the SAF. Finally, note that those lines end up at the same position between Iceland and Scotland, so in both cases, the surface water included in the SAF (as defined here) feeds the Atlantic inflow

draft

to the Nordic Seas.

To conclude, it seems that we were wrong in suggesting an eddy blocking of the northern branch. We now see that the frontal zone moved to the south, leaving no transport for the northern branch identified during the previous decade.

Note that Fig. 2 will not be included in the new ms.

Furthermore, as the authors mention (lines 520-527), they expected an expansion of the SPG. Could you better discuss what the displacement of the SAF actually means?

We agree that this subject requires more discussion, so we will displace the part of the last paragraph of the discussion about the expansion of the SPG to the fourth paragraph of the discussion where we indicated the southeastward displacement of the SAF in 2014. There, we will better explain that the "displacement of the SAF" is the southeastward displacement of the front along the OVIDE section in 2014 as compared with the 2002-2012 average. In fact, Bersch et al. (2002) interpreted that in the eastern SPNA, during the warming period from mid-1990s, there was a northwestward displacement of the SAF coinciding with a contraction of the SPNA. Following Bersch et al. 2002, we proposed that the SAF southeastward displacement suggests a new expansion of the SPNA, consistently with the persistently positive winter NAO index since 2011 (except in 2012). However, both the meandering of the NAC in the eastern SPNA and the lack of distance in time make difficult a strong assessment involving decadal variability as in Bersh et al. (2002).

The green box in Fig. 9 seems very large to me to be considered as the eastern North Atlantic. It is rather peculiar that in the net freshwater field you are averaging over an area that almost symmetrically includes positive and negative net FW.

Surely, the referee is right and the box is somehow large to name it the "eastern subpolar North Atlantic". However, our intention was to define a box containing the whole OVIDE section, including upstream and downstream anomalies. We were also greatly

surprised when we saw the almost symmetrically pattern of the air-sea freshwater flux, with the OVIDE section as the diagonal of the green box separating the negative and positive net FW. It is actually an interesting subject for our future research. It necessarily has a physical explanation, but it can also be fortuitous. In spite of the symmetry, the integrated net FW is positive, which is in agreement with the fact that the eastern SPNA is getting fresher. Diminishing the box (to the northeast) reinforces our conclusion.

The eddy part of the paper is clear, although full of details, it completes the picture well. It is however not easy from a visualization point view to see the eddies and the colors in the figure. Suggest to improve this and make it as clear as possible to the readers, perhaps similar to Fig. 6 of D2016.

We understand your concern. We did different figures representing AVISO information (ADT, and surface velocity) when preparing the first version of the manuscript, and we thought that the representation we propose is more intuitive for our colleagues in biogeochemistry who are interested in the velocity information. We plotted the same way than D2016 but even if the eddies are visible, reading the ADT contours requires some skills typical of the physical community. Therefore, we prefer to keep our presentation, but, to guide the reader, we will introduce more information about the colors of the squares in the next version of the ms.

Consider adding a reference to a recent paper by Rossby et al. (2017, JGR) on the fluxes across 59.5N. Their MOC transport estimate is in line with yours.

Thank you, we will add this reference to reinforce our result.

Minor comments

Please replace 'Hydrological' with 'Hydrographic' throughout.

Ok, totally agree.

Figure 1 is busy and therefore making an effort to explain all the signs is important. For example, you should indicate what the stars represent early on. You may also want to

add the names of the different NAC branches here.

Yes, we agree Figure 1 is busy and with a lot of information necessary to understand the paper. The meaning of the stars is already indicated in the Figure caption of Fig. 1. We will add NNAC, SAF, SNAC and IC (Irminger C.) in the figure.

lines 251: Please keep it consistent with the decimal throughout the paper.

Ok, thank you, we will add one decimal to be consistent throughout the paper.

The bathymetry can hardly be seen in the AVISO figures.

Right, we will draw the bathymetry with thicker gray lines.

lines 505: Define the SPG acronym. And no need for the SPG acronym in the last paragraph of the discussion.

Right, in any case that part of the discussion will be rewritten, and the definition of the SPG will be done in the introduction.

Caption Fig. 9; It is not clear in the text that the anomalies are for 2014.

The referee is right, we will explicitly indicate that it is the 2014 anomaly.
* * *
[Figure]

Figure 1: Heat content anomalies in relation to the mean heat content for the period 2002 - 2012 in the upper 1000m of the region 40°N-60°N and 45°W-10°W. Grey line is the monthly time series; black line is the 2-year running mean of the monthly time series. Data source: EN4 database (Good et al. 2013).

**Fig. 1.**

[Figure]

Fig. 2: Contours of the Absolute Dynamical Topography averaged over 2002-2012 (in black and grey), after removing the overall trend of 2.8mm/yr. Contours are every 0.05m. Thick contours correspond to the levels encompassing the SAF front during OVIDE cruises.

**Fig. 2.**

[Figure]

Figure 2: Contours of the Absolute Dynamical Topography averaged over 2014 (in thin lines). Contours are every 0.05m. Thick contours correspond to the levels encompassing the SAF front during OVIDE cruises: bold red lines for the mean 2002 – 2012 and bold black lines for 2014. Note that the temporal trend on the mean ADT over the whole box (2.8mm/yr) was removed. Bathymetry (1000m step contours) and the OVIDE section are plotted in white. Colors represent the absolute velocity of the current (yellow for velocities stronger than 0.3m/s). This figure will be added to the new ms.

**Fig. 3.**

---

## Author Response (AR2)

Dear Gilles,

Thank you very much for your comments.

We have modified the 2 lines you proposed:

1."In agreement with Grist et al. (2015), we found that the air-sea heat flux is responsible for **most of** the cooling observed in the surface-intermediate layers. "

2. "Dutchez" by **"Duchez".**

We have carefully revised all the abbreviations in the reference list. Concerning Frakjan-Williams et al., 2017, it was published in Scientific Reports (Sci. Rep.).

All the best,

Patricia Zunino

[revised manuscript text omitted]